# Extending the defect tolerance of halide perovskite nanocrystals to hot carrier cooling dynamics

Junzhi Ye [1,2,10], Navendu Mondal [3,10] ✉, Ben P. Carwithen [3], Yunwei Zhang[4], Linjie Dai [1,5], Xiang-Bing Fan[6], Jian Mao [5,7], Zhiqiang Cui[4], Pratyush Ghosh [1], Clara Otero-Martínez[8], Lars van Turnhout [1], Yi-Teng Huang [2], Zhongzheng Yu [1], Ziming Chen [3], Neil C. Greenham [1], Samuel D. Stranks [1,5], Lakshminarayana Polavarapu[8], Artem Bakulin [3], Akshay Rao [1] & Robert L. Z. Hoye [2,9] ✉

Defect tolerance is a critical enabling factor for efficient lead-halide perovskite materials, but the current understanding is primarily on band-edge (cold) carriers, with significant debate over whether hot carriers can also exhibit defect tolerance. Here, this important gap in the field is addressed by investigating how intentionally-introduced traps affect hot carrier relaxation in $CsPbX_3$ nanocrystals (X = Br, I, or mixture). Using femtosecond interband and intraband spectroscopy, along with energy-dependent photoluminescence measurements and kinetic modelling, it is found that hot carriers are not universally defect tolerant in $CsPbX_3$, but are strongly correlated to the defect tolerance of cold carriers, requiring shallow traps to be present (as in $CsPbI_3$). It is found that hot carriers are directly captured by traps, instead of going through an intermediate cold carrier, and deeper traps cause faster hot carrier cooling, reducing the effects of the hot phonon bottleneck and Auger reheating. This work provides important insights into how defects influence hot carriers, which will be important for designing materials for hot carrier solar cells, multiexciton generation, and optical gain media.

The power conversion efficiency (PCE) of lead iodide-based perovskite photovoltaics (PVs) has now reached a certified value of 26.7% under 1-sun illumination[1], which is rapidly approaching the radiative PCE limit of ~30%[2,3]. This radiative limit in efficiency comes from the inefficient management of heat dissipation, especially for charge-carriers excited above the bandgap (i.e., hot carriers, or HCs)[4-9]. However, practically, the excess energy of the HCs cannot be fully utilized and extracted efficiently because HC cooling is much faster than charge-carrier extraction[9-11]. Therefore, understanding the origin of the fast-cooling process in halide perovskites with different bandgaps and

[1]Cavendish Laboratory, University of Cambridge, Cambridge, UK. [2]Inorganic Chemistry Laboratory, University of Oxford, Oxford, UK. [3]Department of Chemistry and Centre for Processable Electronics, Imperial College London, Molecular Sciences Research Hub, London, UK. [4]School of Physics, Sun Yat-sen University, Guangzhou, China. [5]Department of Chemical Engineering and Biotechnology, University of Cambridge, Cambridge, UK. [6]Department of Engineering, University of Cambridge, 9 JJ Thomson Avenue, Cambridge, UK. [7]State Key Laboratory of Photovoltaic Science and Technology, Shanghai Frontiers Science Research Base of Intelligent Optoelectronics and Perception, Institute of Optoelectronics, Fudan University, Shanghai, China. [8]CINBIO, Universidade de Vigo, Materials Chemistry and Physics Group, Department of Physical Chemistry, Campus Universitario As Lagoas, Marcosende, Vigo, Spain. [9]Department of Materials, Imperial College London, London, UK. [10]These authors contributed equally: Junzhi Ye, Navendu Mondal. ✉e-mail: n.mondal@imperial.ac.uk; robert.hoye@chem.ox.ac.uk

compositions is crucial for developing approaches to sufficiently slow down HC cooling, such that these HCs can be collected. In general, HCs release this excess energy through ultrafast (<100 fs) carrier-carrier scattering[12–14], followed by carrier-phonon interaction events (<1 ps) to reach the lattice temperature[10,15–18]. Moreover, studying HC dynamics provides a fundamental understanding of the carrier-carrier and carrier-phonon interactions in these materials with implications for charge-carrier transport and material stability[19,20].

One of the key enabling, but unusual, properties of lead halide perovskites (LHPs) is their reported defect tolerance, meaning that the lifetimes and mobilities of free charge-carriers are relatively insensitive to the presence of defects[21–24]. However, defect tolerance has generally been discussed with regard to band-edge (i.e., "cold") carriers undergoing non-radiative recombination, and is typically quantified by the photoluminescence quantum yield (PLQY) and time-resolved photoluminescence (PL) measurements. A critical question is whether this defect tolerance extends to HCs. Bonn et al. suggested this to be so[25], whereas Jiang et al. suggested that while band-edge carriers in MAPbI$_3$ were defect-tolerant, the HC lifetime was shortened due to trapping at the grain boundaries of polycrystalline films[26]. HC-activated trapping[27,28], along with passivation-induced enhancement of HC lifetime[9,15,19,28–31], pressure-induced lattice compression[32] and transport[33], was also observed in a few other instances that can influence the HC cooling. Righetto et al. reported that the HC cooling process can be directly affected by defect trapping on the surface of perovskite nanocrystals (NCs)[28]. Zhou et al. utilized density functional theory (DFT) calculations to suggest that common point-defects, such as iodide and methylammonium vacancies in MAPbI$_3$ perovskite, can adversely affect the HC lifetime[9]. The defect tolerance of HCs is therefore not known, with contradictory evidence in the current literature. Our work aims to address this by systematically establishing the relationship between HC cooling lifetime and defect densities and energies in compositionally-tunable perovskites. This will be important for developing wide-gap and narrow-gap perovskite absorbers for single-junction and tandem solar cells, and could show whether HC perovskite solar cells are attainable[34,35].

To elucidate the role of traps on HC lifetime, CsPbX$_3$ NCs were selected as the material system of interest rather than LHP thin films. This is because of the capability of introducing different defect densities intentionally by controlling the NC surface chemistry through multiple antisolvent purification steps[36] and also its wide applications in display, optical communications and energy harvesting[37–40]. This enables a more direct correlation between the effects of defects and non-radiative recombination, which affects the PLQY. By progressively increasing the defect density, we employed femtosecond pump-probe (PP) and pump-push-probe (PPP) transient absorption (TA) spectroscopy to investigate the effect of trap density and energy on the HC dynamics. The TA spectroscopy and kinetic modelling demonstrate that HC lifetime is governed by both defect density and energy, which are in turn controlled by composition and bandgap. We find that HCs are protected in narrow-bandgap perovskite NCs comprising shallow traps compared to those in wide-gap NCs. By demonstrating the defect-tolerant nature of HCs in halide perovskites, design guidelines are provided that can pave the way for efforts to develop HC solar cells.

## Results
### Intentionally tuning trap densities, and the effects on cold carriers

We synthesised colloidally stable CsPbX$_3$ (X = Br, I) NCs by hot-injection (see Methods for details)[36,41]. To intentionally introduce different defect densities, the as-synthesised CsPbBr$_3$, CsPbBr$_x$I$_{3-x}$ and CsPbI$_3$ NCs were purified multiple times using the low-polarity antisolvent methyl acetate. This process results in the partial removal of surface ligands and halides without affecting the size and structure of the NCs, as shown in Fig. 1a and Supplementary Fig. 1[36,42]. From X-ray

photoelectron spectroscopy (XPS) measurements (Supplementary Fig. 2a–c), there was a decrease in the surface halide to Pb ratio with an increase in the number of purification steps, indicating an increase in the density of surface halide vacancies. The Pb to halide ratio is calculated by our previously-reported method[36], which is based on the integrated area ratio between the Pb $4f$ and I or Br $3d$ core level XPS spectra shown in Supplementary Fig. 3. Following purification, the absorption and PL spectra of the pure halide NCs did not exhibit any significant changes (Fig. 1b, d), while the mixed-halide perovskite NCs significantly blue-shifted (Fig. 1c). This was due to the selective etching (expulsion) of surface iodine species, resulting in bandgap widening[36].

The decrease in the PLQY (Fig. 1) and PL lifetime (Supplementary Fig. 4a, b) of the mixed-halide and pure-bromide perovskites with an increase in the number of purification steps (Fig. 1a) is consistent with the introduction of traps due to halide vacancies[42–44]. The increase in defect density was further confirmed by photothermal deflection spectroscopy (PDS), which shows an increase in the sub-bandgap absorption and Urbach energy as the NCs were washed multiple times with polar antisolvents (Supplementary Fig. 2d–f). By contrast, there was very little change in the PLQY and PL lifetime of the narrow-gap CsPbI$_3$ (Fig. 1a, Supplementary Fig. 4c), despite substantial changes in surface composition (Supplementary Fig. 2c) and sub-bandgap absorbance (Supplementary Fig. 2f) after purification. This is consistent with the greater defect tolerance of CsPbI$_3$, due to the dominant iodide vacancies being shallow.

To gather preliminary insight into the role of defects on the dynamics of HCs, we first performed excitation-energy-dependent PLQY measurements. These measurements (Fig. 1e–g) show that for less defective NCs, the PLQY remained relatively constant with changes in excitation energy. By contrast, for more defective (i.e., doubly purified) CsPbBr$_3$ and CsPbBr$_x$I$_{3-x}$ NCs, the PLQY decreased by ~15% with an excess energy of ~1 eV (Fig. 1e, f). If the PLQY were solely dependent on the recombination of band-edge carriers, we would expect the PLQY to remain the same irrespective of the excitation energy. Indeed, we observe only a small change in PLQY with increasing excess energy in CsPbI$_3$ NCs, and this was not exacerbated by introducing a higher defect density (Fig. 1g), due to the shallow nature of the traps and the defect tolerance of I-based NCs, which is consistent with previous literature[43,45]. Together, these results point to the conclusion that when the excitation energy is high (w.r.t. the bandgap) in wide-gap defective perovskites, HCs undergo additional trap-assisted non-radiative recombination pathways, thus lowering PLQY.

We now utilize a simplified HC trapping model originally developed by Righetto et al.[28] (detailed in Supplementary Note 1 and Supplementary Fig. 5) to explain the HC trapping process. These fits are shown as the dotted line in Fig. 1e–g, and further details on the model are described in Supplementary Note 1 and Supplementary Fig. 5, with fitting parameters shown in Supplementary Table 1. The fitted values suggest that the high defect-density samples tend to have lower radiative recombination constants and greater electronic coupling to traps than the low defect-density samples, and this effect is more significant for wider-gap systems. The apparent 'tolerance' of HCs to traps in CsPbI$_3$ could be due to these traps being shallower than in CsPbBr$_3$ and CsPbBr$_x$I$_{3-x}$ NCs, such that the overlap between the conduction band and trap states is smaller with a smaller energy offset for the HCs to be trapped[28]. Consistent with this proposed explanation, we verified from DFT calculations that the halide vacancy in CsPbI$_3$ (0.278 eV from the conduction band minimum) is much shallower compared to Br/I-based (0.513 eV) and Br-based (0.666 eV) NCs (Supplementary Fig. 6). The absolute values of these trap states might not be completely accurate, but the trend in defect positions match with the literature[43,46].

Following this qualitative evidence, we now turn our attention to characterizing the effects of defects through time-resolved

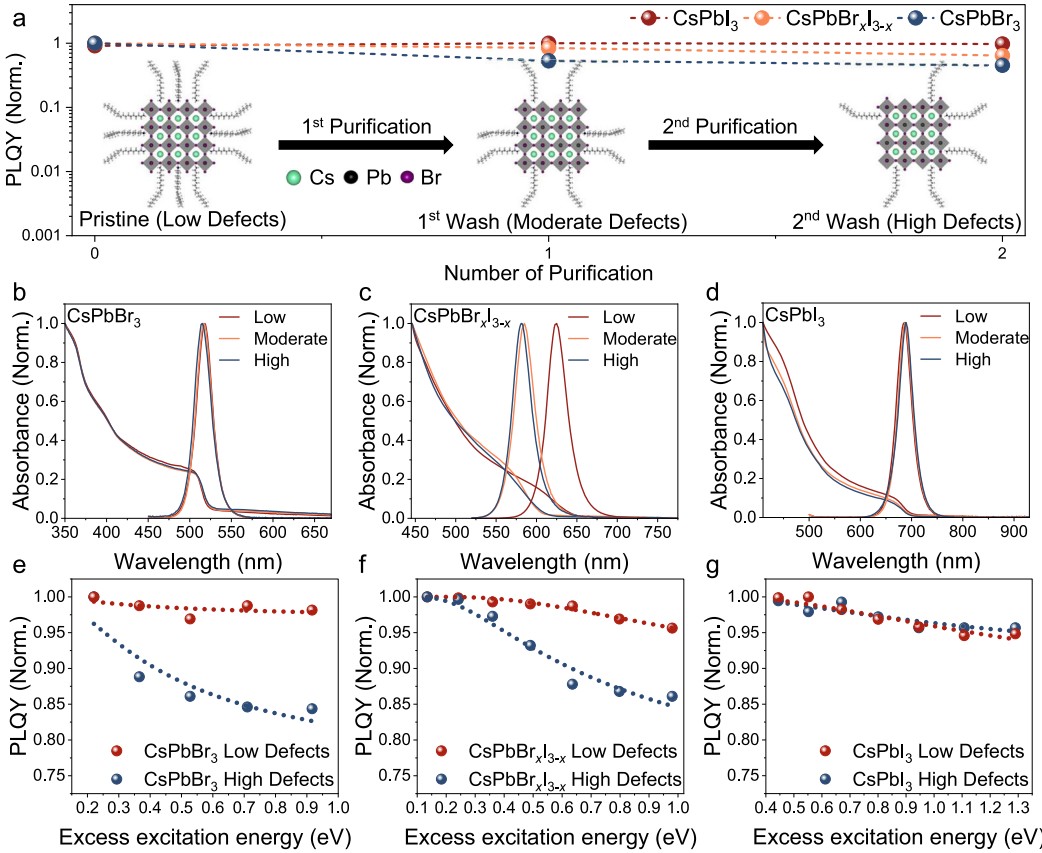

**Fig. 1 | Effect of intentionally introduced defect states on the optoelectronic properties of CsPbX₃ nanocrystals (X = I, Br, or I/Br mixture). a** Change in the normalized photoluminescence quantum yield (PLQY) of $CsPbI_3$ (red), $CsPbBr_xI_{3-x}$ (orange) and $CsPbBr_3$ (blue) nanocrystals after successive purification steps. All PLQY values were normalized to the PLQY of the pristine samples. Illustrations inset show the changes to the surface defect and ligand density of the nanocrystals before and after each purification step. Comparison of the photoluminescence (PL) and absorption spectra of pristine (low defect density), singly purified (moderate defect density), and doubly purified (high defect density) **b** $CsPbBr_3$, **c** $CsPbBr_xI_{3-x}$

and **d** $CsPbI_3$ nanocrystals. Excitation-wavelength-dependent PLQY for **e** $CsPbBr_3$, **f** $CsPbBr_xI_{3-x}$, and **g** $CsPbI_3$ nanocrystals. All PLQY values were normalized to the PLQY measured with the excitation laser energy that is closest to the bandgap of the materials. In each case, a comparison is made between nanocrystals with low (red) and high (blue) defect densities. The model fit to the excitation-dependent measurements was obtained from ref. 28. The PLQY and PL lifetime measurements here, along with the XPS measurements (Supplementary Figs. 2 and 3), suggest that we have successfully prepared NCs with different defect densities.

spectroscopic techniques, namely two-pulse TA spectroscopy. Figure 2a, b shows the TA spectra (within 0.5–30 ps, note labelling of the vertical axes as Δ*T/T*) of the low and high defect density $CsPbBr_3$ samples. Three main spectral signatures are often observed in perovskite TA spectra: (i) a high-energy, broad negative photo-induced absorption (PIA), usually caused by a refractive index change, but can also be ascribed to polaronic signatures[47], (ii) positive ground state bleach (GSB), attributed to the depopulation of band-edge carriers, and (iii) a negative sub-bandgap short-lived PIA, which is often ascribed to band-gap renormalization (BGR). The early red-shift of the GSB (feature ii) over time is an indication of hot carriers with excess energy relaxing to the band-edge (steady-state absorption onset). The decay of the sub-gap PIA is correlated with the rise of the GSB (due to state-filling during HC cooling). As we have intentionally introduced defects into the NCs, the sub-bandgap region also has a positive trap bleach (TB) signature overlaid with the negative BGR feature. Figure 2c displays the sub-bandgap kinetics (at 535–545 nm) which show a clear difference between the low (red dots) and high (blue dots) defect density $CsPbBr_3$ NCs. While initially negative due to BGR upon excitation, Δ*T/T* then evolves into a positive TB feature within 1 ps, indicating carriers are being trapped into the defect states even during the HC cooling time window. The TB subsequently decays over ~100 ps. Similarly, the sub-gap kinetics are also dependent on defect density in the mixed-halide NCs (Fig. 2d). However, no obvious change is

observed for pure I-based NCs (Fig. 2e). To make a careful assessment of the sub-band-gap region in the TA signal, which comprises of a convolution between the PIA and TB, we performed singular value decomposition analysis (results shown in Supplementary Fig. 7), which clearly resolves their individual contributions and is consistent with our analysis presented in Fig. 2. Interestingly, a positive Δ*T/T* component is identified after deconvolution for the high defect density $CsPbBr_xI_{3-x}$ NCs due to trap bleaching, but which is not apparent for the $CsPbI_3$ NCs (Supplementary Fig. 7). This is consistent with our energy-dependent PLQY measurements, suggesting that HCs in I-based NCs are less influenced by defects.

## Influence of traps on hot carrier cooling lifetime probed by two-pulse transient absorption spectroscopy

Both excitation-energy-dependent PLQY and the sub-bandgap states of the TA analysis suggest that defects can influence the HC cooling process for wide-bandgap systems based on indirect monitoring of the band-edge carriers after cooling. In this section, we will directly measure and discuss the effect of defects on the HC cooling lifetime ($\tau_{cool}$), which is defined as the time required for HCs to equilibrate with the surrounding lattice temperature, using two-pulse TA spectroscopy. Figure 3a–f shows the 2D colour map of the TA spectra for all materials systems with increasing defect densities measured at high fluence. It is clearly evident that the GSB signals of the $CsPbBr_3$ and $CsPbBr_xI_{3-x}$ NCs

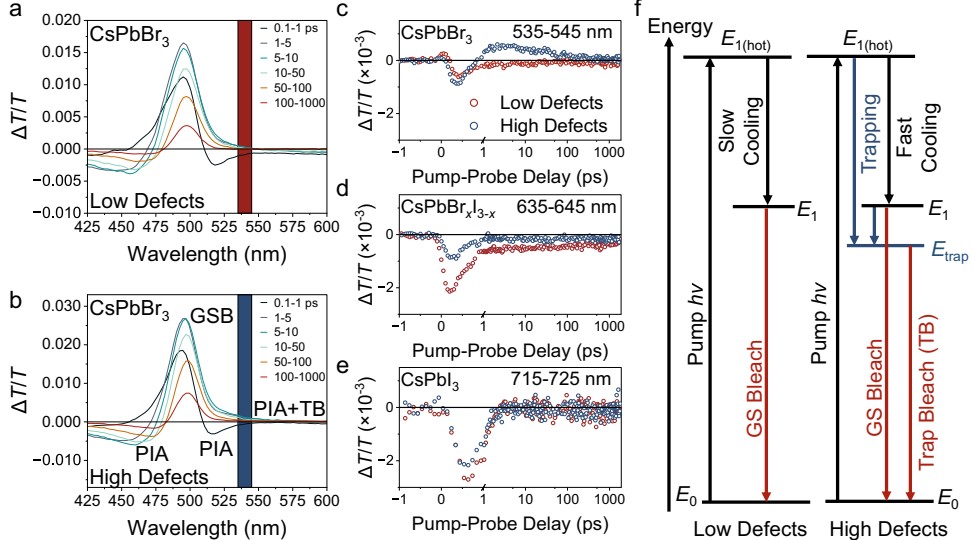

**Fig. 2 | Evidence of carrier trapping from pump-probe transient absorption spectroscopy. a, b** Transient absorption spectra of singly-purified (low defect density) and doubly-purified CsPbBr₃ NCs (high defect density). Red and blue regions indicate the sub-bandgap regions (containing both trap bleach (TB) and photo-induced absorption (PIA); 535–545 nm) that were integrated to determine the kinetics. The kinetics of sub-bandgap regions for singly (low defect density)- and double-purified (high defect density) **c** CsPbBr₃ NCs (probed at 535–545 nm), **d** CsPbBrₓI₃₋ₓ NCs (probed at 635–645 nm), and **e** CsPbI₃ NCs (probed at

715–725 nm). The TA measurements were performed at 116.6 μJ cm⁻², and the spectra are shown in Supplementary Figs. 8–10. TB was extracted from the measured spectra through singular-variable decomposition (SVD), shown in Supplementary Fig. 7. **f** Schematic representation of the charge-carrier relaxation processes highlighting the carrier trapping events, with energy levels indicated here arbitrarily. The NC solution was pumped with photon energies significantly higher than the bandgap ($\hbar\omega_{\text{pump}} = 3.1$ eV).

become narrower with an increase in the defect densities in these systems. The narrowing of the GSB is an indication of fast HC cooling. However, the GSB of CsPbI₃ NCs did not show any noticeable narrowing as we increased the defect density (Fig. 3g–i).

To further investigate the trapping effect on the kinetics of HC cooling, we extracted cooling temperature and lifetimes via the widely used procedure of fitting a Maxwell–Boltzmann distribution (Eq. 1) to the high-energy GSB tail, by approximating it from the Fermi-Dirac distribution of HCs[10,48] (Eq. 1), as shown in Supplementary Figs. 8–10:

$$\frac{\triangle T}{T}(E) \propto e^{-\frac{E-E_{\text{F}}}{k_{\text{B}}T_{\text{c}}}} \tag{1}$$

where $\frac{\triangle T}{T}(E)$ is the normalized differential transmittance obtained from the TA measurement, $T_{\text{c}}$ is the carrier temperature, $E$ is carrier energy and $E_{\text{F}}$ the Fermi level. The energy loss rate ($J_{\text{loss}}$) in eV ps⁻¹ can be estimated for NCs with different compositions and defect densities based on Eq. 2[49]:

$$J_{\text{loss}} = -\frac{3}{2}k_{\text{B}}\frac{dT_{\text{c}}}{dt} \tag{2}$$

The results are shown in Fig. 3j–l. For Br-based NCs, $J_{\text{loss}}$ increased from around 0.03 eV ps⁻¹ to 0.2 eV ps⁻¹ upon defect introduction at a carrier temperature of 1000 K (Fig. 3j). Similarly, $J_{\text{loss}}$ increased from around 0.008 eV ps⁻¹ to 0.04 eV ps⁻¹ for low to moderate defect densities, and to 0.17 eV ps⁻¹ for high defect densities in Br/I-based NCs at 1000 K (Fig. 3k). In contrast, $J_{\text{loss}}$ remained relatively constant for I-based NCs, which is around 0.13 eV ps⁻¹ at 1000 K. This suggests that the presence of defects introduces an additional energy loss pathway in wider bandgap NCs, and thus the loss rate is higher for high defect density systems. However, these defects are less influential for the low-gap CsPbI₃, since the energy loss rate did not change drastically for these NCs.

The extracted charge-carrier temperatures as a function of time delay for different initial carrier densities or fluences for NCs with

different compositions are presented in Supplementary Fig. 8 to 10h–g. For low fluence, the time-dependent changes in carrier temperature can be fitted with a single exponential decay, as shown in Supplementary Tables 2–4, corresponding to rapid carrier cooling via Fröhlich interactions, in which scattering between charge-carriers and longitudinal optical (LO) phonons is the dominant pathway at room temperature[10]. At moderate excitation density, the HC lifetime gradually increases as carriers compete for a finite availability of phonon modes into which they may transfer their excess energy—a phenomenon known as the hot phonon bottleneck[50,51]. Apart from phonon heating, enhanced carrier-carrier interactions at high carrier concentrations could also contribute to prolonged HC lifetimes through the non-radiative Auger heating process[10,11]. Here, the energy released upon recombination of an electron-hole pair is transferred to a neighbouring carrier, thus replenishing the population of HCs. Unlike bulk systems, this Coulombic interaction-mediated Auger heating process often dominates in quantum confined NCs. Thus, at high carrier density, the appearance of an additional slower component after the initial sub-ps decay is assigned to Auger heating in our NC-based systems[10,11]. As a result, biexponential fitting is required to capture the whole cooling trajectory, in which the fast and slow lifetime components (under moderate to high carrier densities) can be ascribed to the hot phonon bottleneck and Auger heating effects, respectively, and the fitted results are listed in Supplementary Tables 2–4. Interestingly, this second slow cooling component at high fluence became less obvious for the moderately-defective and highly-defective NCs. To give an example, for Br-based systems, at a carrier density of $12.5 \times 10^{17}$ cm⁻³, the HC cooling lifetime decreases from 3.8 ± 0.3 ps (for the low defect density sample), to 2.6 ± 0.4 ps and 1.0 ± 0.1 ps for the moderate and high defect density samples, respectively (Fig. 4a and Supplementary Table 2), due to the substantial decrease in the $t_2$ component, which is strongly influenced by excitation fluence. This indicates that the presence of higher defect densities introduces additional relaxation routes (i.e., trap-mediated) to the intrinsic intra-band relaxation process, therefore leading to a reduction in the cooling time over the entire carrier density range studied here. The same

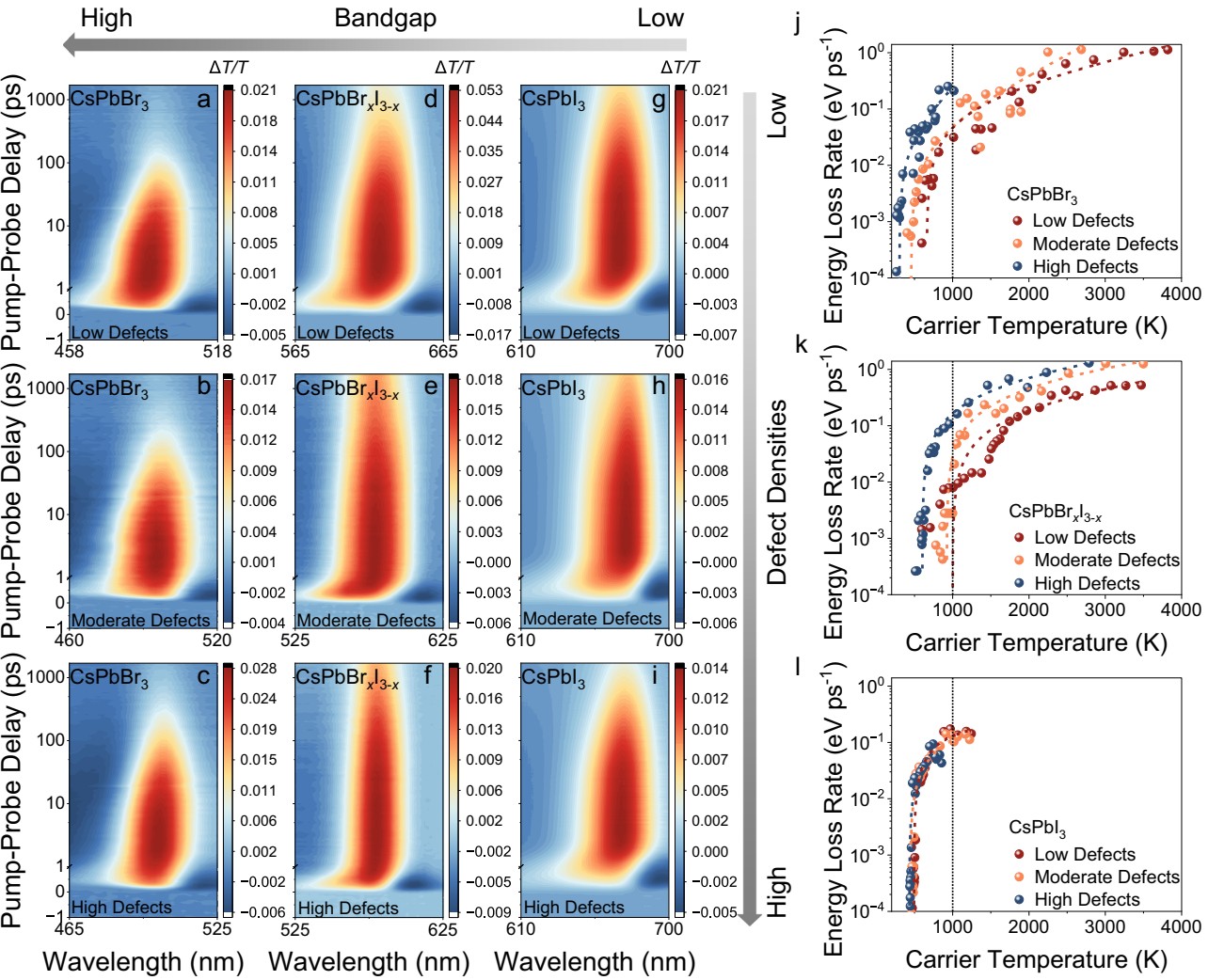

**Fig. 3 | Hot carrier energy loss rate based on pump-probe transient absorption spectroscopy measurements.** TA maps for colloidal solutions of low, moderate and high defect density **a–c**, CsPbBr$_3$, **d–f**, CsPbBr$_x$I$_{3-x}$ and **g–i** CsPbI$_3$ perovskite nanocrystals under 400 nm wavelength excitation. Energy loss rate for different defect concentrations in **j** CsPbBr$_3$, **k** CsPbBr$_x$I$_{3-x}$, and **l** CsPbI$_3$ NCs. A vertical dotted line is placed at 1000 K carrier temperature ($T_c$) to visually aid the direct comparison between pristine (low defect density, red), singly purified (moderate defect density, yellow) and doubly purified NCs (high defect density, blue). The energy loss rate is based on the relaxation lifetime at 194.3 μJ cm$^{-2}$ for CsPbBr$_3$ (carrier density of 12.5 × 10$^{17}$ cm$^{-3}$), 178.2 μJ cm$^{-2}$ for CsPbBr$_x$I$_{3-x}$ (carrier density of 43.5×10$^{17}$ cm$^{-3}$) and 193.4 μJ cm$^{-2}$ for CsPbI$_3$ (carrier density of 16.6×10$^{17}$ cm$^{-3}$).

results are shown in Supplementary Fig. 9h–g for CsPbBr$_x$I$_{3-x}$ NCs, where the second component appeared clearly at a carrier density of 58.0 × 10$^{17}$ cm$^{-3}$ or higher, and almost disappeared for the moderate and high defect density samples. The cooling lifetime decreased from 7.1 ± 0.1 ps (for the low defect density sample) to 4.8 ± 0.1 ps to 4.5 ± 0.1 ps for the moderate to high defect density sample (Fig. 4b and Supplementary Table 3). However, for the CsPbI$_3$ NCs, the change of the spectra, map and extracted carrier cooling kinetics did not vary significantly with the number of purification steps (Supplementary Fig. 10). The HC lifetime was found to be similar between the pristine and defective samples (around 12–13 ps at the carrier density of 33.3 × 10$^{17}$ cm$^{-3}$, as shown in Fig. 4c and Supplementary Table 4). In summary, for wider bandgap systems (Br and Br/I-based NCs), $\tau_{cool}$ is reduced as the defect density increased, but this phenomenon is less obvious for lower bandgap I-based NCs (Fig. 4d). We also observed that the differences in the extracted HC lifetimes for samples with different defect densities become smaller as the carrier density increases due to the trap filling effect. At higher carrier density (generated through excitation with high fluence), traps are occupied, and the additional loss pathway for the HC is blocked, such that the HC lifetimes tend to approach similar values for NCs with different defect densities, as

shown in Fig. 4b. This effect of trap filling at higher excitation fluence is further elaborated later in this work.

## Influence of traps on hot carrier cooling lifetime probed by three-pulse transient absorption spectroscopy

In the two-pulse PP approach described thus far, all charge-carriers (hot and cold) are formed by a single ('pump') excitation event. As such, the density-dependence of the intrinsic HC lifetime (including the effect of trapping) can be obscured by 'Auger re-heating' effects at high excitation fluences. Moreover, in PP-TA spectroscopy, the HCs are initially formed with different excess energies due to the variations in bandgap between different NC compositions. Therefore, to obtain a more quantitative description of the effect of traps, we employed a three-pulse pump-push-probe (PPP) spectroscopic approach, in which an additional narrowband near-IR push pulse is introduced after the pump to optically re-excite the band-edge (cold) carriers to a higher energy (hot) state. The working principles of the PPP-TA setup is shown schematically in Fig. 5a, with specific details in Methods. A 1300 nm push wavelength is chosen, as this corresponds to a broad featureless intraband PIA for these systems, as shown in Supplementary Fig. 11, and consistent with previously described literature[52,53]. To

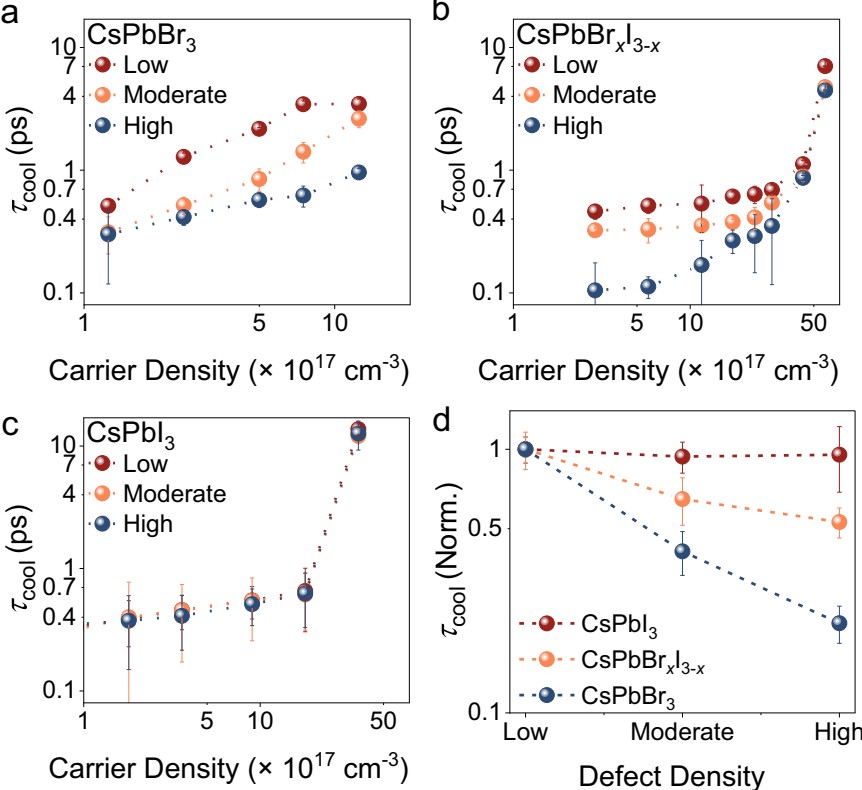

**Fig. 4 | Hot carrier cooling kinetics based on pump-probe transient absorption spectroscopy.** HC lifetime ($\tau_{cool}$) obtained by fitting pump-probe transient absorption spectroscopy measurements of **a** CsPbBr$_3$, **b** CsPbBr$_x$I$_{3-x}$, and **c** CsPbI$_3$ NCs with low (red), yellow (moderate) and high (blue) defect densities. **d** Hot carrier lifetime against defect density of CsPbI$_3$ (red), CsPbBr$_x$I$_{3-x}$ (yellow) and CsPbBr$_3$ (blue) perovskite NCs. The normalized $\tau_{cool}$ for CsPbBr$_3$ is based on the relaxation lifetime at 194.3 μJ cm$^{-2}$ fluence (carrier density of 12.5 × 10$^{17}$ cm$^{-3}$), for CsPbBr$_x$I$_{3-x}$ at 178.2 μJ cm$^{-2}$ fluence (carrier density of 43.5 × 10$^{17}$ cm$^{-3}$), and for CsPbI$_3$ is at 193.4 μJ cm$^{-2}$ fluence (carrier density of 16.6 × 10$^{17}$ cm$^{-3}$). Error bars represent uncertainty in numerical fitting of the cooling lifetime at each carrier density.

ensure the push exclusively acts on cold carriers and the resulting excess energy is the same across all systems irrespective of their band gap, we introduced the push beam 10 ps after the initial 400 nm pump excitation, which is when all hot carriers have relaxed to the band-edge.

To verify our hypothesis that the near-IR push pulse exclusively acts to create a transient hot state, we first discuss its effect on the broadband visible TA signals of CsPbBr$_3$ NCs. As can be seen from Supplementary Fig. 12, notable differences in all TA features are observed immediately following the arrival of the push: the magnitudes of the GSB and above-gap PIA are momentarily reduced, concomitant with the reappearance of the sub-gap PIA. Each of these features are also seen upon above-gap excitation in the above two-pulse experiments.

Representative GSB kinetics obtained for CsPbBr$_3$ NCs under the same pump fluence in both PP and PPP approaches are displayed in Fig. 5b, showing the depletion and subsequent recovery of the GSB due to the push pulse. As shown in Fig. 5c, increasing the push fluence increases the density of HCs formed, and therefore also the magnitude of this initial change ($\delta\Delta$OD) in the GSB. Fitting the GSB recovery with a Gaussian-convolved mono-exponential decay yields the HC lifetime ($\tau_{hot}$). A low pump fluence is used in all PPP measurements (such that the average carrier density per NC $\langle N \rangle \ll 1$), and the HC lifetimes extracted as a function of initial HC density ($n_{hot,0}$). That is, the hot phonon bottleneck effect can be isolated within a total carrier density that precludes competing Auger re-heating effects. We note that identical HC lifetimes are extracted from the kinetics of the GSB and both PIAs (Supplementary Fig. 12), but the GSB is analysed in all cases hereinafter due to its greater signal amplitude.

The HC lifetime for all three halide compositions, as well as their respective purified samples, are plotted as a function of HC density in Fig. 5d–f. The fitted HC lifetimes obtained from PPP-TA follow similar trends to the PP-TA results described earlier. In all cases, there is a general increase in $\tau_{hot}$ with increasing HC density, which is consistent with the hot phonon bottleneck effect.

Although the general trend of an increase in HC lifetime with increasing HC density is observed across all the NC compositions investigated, systematic differences are observed between the pristine and defective samples. We rationalize this observation as follows. It is clearly evident from Fig. 5d, e that the intrinsic HC lifetimes for the defective Br- and Br/I-based NCs are shorter than their respective pristine samples. This points to halide vacancy-related defects in CsPbBr$_3$ and CsPbBr$_x$I$_{3-x}$ systems introducing a competing channel for the intraband relaxation process, leading to faster dynamics. Under the highest density of HCs, the differences in HC lifetime between pristine and defective samples are diminished. This likely arises due to trap state saturation: when all of the vacant trap sites are filled, the competing pathway for intraband relaxation processes is blocked and therefore the HC lifetime approaches that of their respective pristine samples. In the CsPbI$_3$ NCs, no differences are observed between the pristine and defective samples at any HC density in spite of the introduction of halide vacancies apparent from compositional analysis (Supplementary Fig. 2a−c), confirming that defect tolerance is retained for HCs in this system.

To illustrate and quantitatively formulate the above hypothesis regarding the various HC relaxation routes, particularly carrier-trap and carrier-phonon interactions, a simple kinetic model was

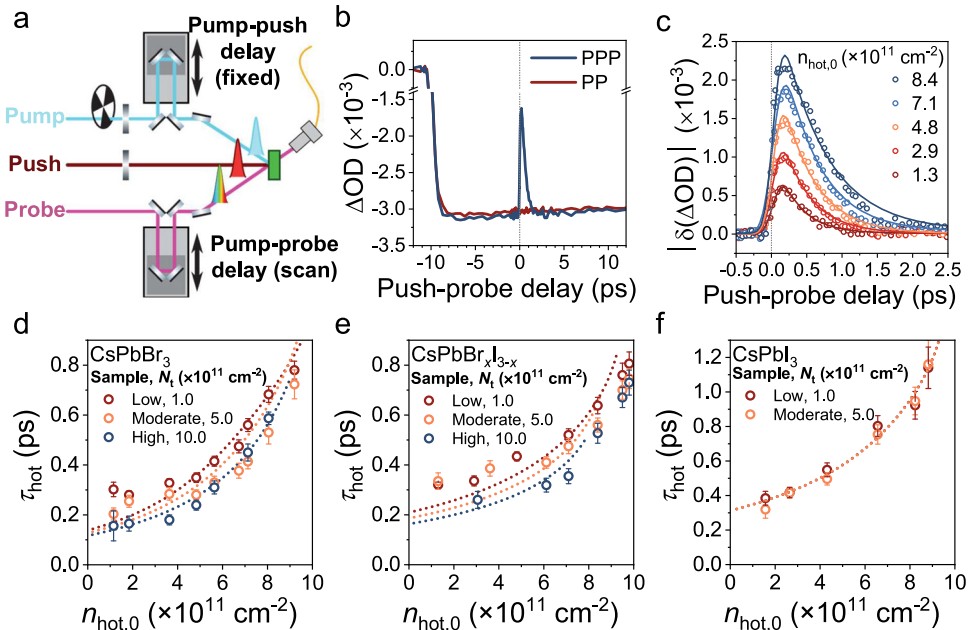

**Fig. 5 | Hot carrier cooling kinetics based on pump-push-probe transient absorption spectroscopy. a** Schematic diagram of the pump-push-probe TA setup used. **b** Exemplar comparison of the pump-probe (PP, dark red) and pump-push-probe (PPP, blue) GSB decay kinetics with a pump-push delay of 10 ps for low defect density CsPbBr$_3$ NC solution. **c** Hot carrier density (or push-fluence) dependent representative GSB decay curves. HC lifetimes obtained from fitting the PPP TA measurements for **d** CsPbBr$_3$, **e** CsPbBr$_x$I$_{3-x}$, and **f** CsPbI$_3$ NCs, respectively, with low (red), moderate (yellow) and high (blue) defect densities. Dotted lines are results from the kinetic model described in final sub-section of Results. Error bars represent uncertainty in numerical fitting of the cooling lifetime at each carrier density.

developed (Eq. 3):

$$\frac{dn_{hot}}{dt} = \widetilde{I}(t)n_{cold} - \alpha n_{cold}n_{hot} - \phi n_{ph}n_{hot} - \gamma(N_t - n_t^*)n_{hot} \quad (3)$$

where $\widetilde{I}$ represents the push Gaussian envelope. $n_{hot}$, $n_{cold}$, $n_{ph}$, $N_t$ and $n_t^*$ are the densities of hot and cold carriers, vacant phonons, and total and occupied traps, respectively. The coefficients $\alpha$ and $\phi$ represent hot-cold carrier and carrier-phonon scattering, and are based on values extracted from our previous work[18,54,55], while $\gamma$ denotes the propensity for these defects to act as traps for HCs. The full system of coupled differential equations can be found in Supplementary Eqs. S6–9. The modelled results are overlaid on the experimental data in Fig. 5d–f, and show good agreement with physically-reasonable fitting parameters (Supplementary Table 5). In the modelling, $N_t$ is mainly governed by the number of purification steps. As $N_t$ increases, the cooling lifetime is shortened in the Br-containing NCs, for which $\gamma$ values are non-zero. The effect of defect densities on HC cooling can also be influenced by the push fluence due to the trap filling effect, i.e., as $n_t^* \rightarrow N_t$, the trapping term in Eq. 3 approaches zero. This explains the observed up-curvature in Fig. 5d–f for the purified CsPbBr$_3$ and CsPbBr$_x$I$_{3-x}$ NCs. Meanwhile, $\gamma = 0$ in CsPbI$_3$, leading to negligible differences between purified and pristine NCs. As such, the numerical model compares very satisfactorily with salient trends observed in experimental Fig. 5d–f, giving further credence to the notion that trapping can accelerate HC loss. Concretely, HCs undergo direct trapping – rather than sequential cooling then trapping, which would not manifest as a change in HC lifetime – and the driving force for this process is the energy offset to deep traps in the defect-intolerant NCs. Furthermore, these results highlight more generally that an interplay of the hot phonon bottleneck and trap state saturation could lead to longer HC lifetimes in the defective samples.

## Discussion

In conclusion, we found that the defect tolerance of band-edge carriers in LHPs is a good predictor for that of HCs. HCs are lost due to non-radiative recombination more readily in the wide-gap CsPbBr$_3$ and CsPbBr$_x$I$_{3-x}$ perovskites than in the narrow-gap CsPbI$_3$ perovskites. After intentionally introducing additional defects to the wide-gap NCs, faster HC relaxation is observed due to the reduction of the hot phonon bottleneck and Auger reheating effects. We posit that HCs can therefore trap directly, without needing to first cool to the band-edge. The finding suggests that HCs are not universally defect-tolerant in CsPbX$_3$ perovskites, but rather depend on trap depth, which may help to explain the current conflicting conclusions in the literature. Our work implies that there is a greater chance of realizing HC solar cells or lasing from iodide-based perovskites, and that these devices can tolerate the extra defects that could be generated during the ligand exchange needed to improve the macroscopic charge-carrier transport of the NC film. We expect that these results can be more widely generalized beyond the inorganic halide perovskite system we investigated here towards hybrid systems as well. Our results suggest that narrow-gap formamidinium-based perovskites, found to potentially exhibit greater defect tolerance than methylammonium-based perovskites[56], are promising for further investigation.

## Methods

### Materials

Cesium carbonate (Cs$_2$CO$_3$, 99.9%), formamidine acetate (CH$_4$N$_2$·C$_2$H$_4$O$_2$, 99%), methylammonium bromide (CH$_3$NH$_3$Br, >99%, anhydrous), lead (II) bromide (PbBr$_2$, >98%), lead (II) iodide (PbI$_2$, 99%), 1-octadecene (ODE, C$_{18}$H$_{36}$, 90%) oleic acid (OA, C$_{18}$H$_{34}$O$_2$, 90%) and oleylamine (OLA, C$_{18}$H$_{37}$N, 70%) were purchased from Merck. Methyl acetate (CH$_3$COOCH$_3$, 99%) and toluene (C$_7$H$_8$, <99.8%) was supplied by Alfa Aesar. All chemicals were used without further purification.

### Preparation of a stock of cesium-oleate (Cs-OL)

In a typical synthesis, 407 mg of Cs$_2$CO$_3$ (1.25 mmol) and 1.25 mL of oleic acid (3.5 mmol) were added to 20 mL of 1-octadecene in a 50 mL sample vial. The resulting mixture was heated at 150 °C under stirring until the salt was completely dissolved. The Cs-oleate complex

generally precipitated at room temperature, however, it can be easily dissolved again by continuous stirring at 120 °C.

## Preparation of a stock solution of different lead halides (PbX₂)

In a typical synthesis, 345 mg of $PbBr_2$ (0.94 mmol) or 433 mg of $PbI_2$ (0.94 mmol)) was added to a mixture of 2.5 mL of OLA (5.3 mmol), 2.5 mL of oleic acid (7.0 mmol) and 25 mL of 1-octadecene in a 50 mL sample vial. The resulting mixture was heated at 150 °C under continuous stirring until the salt completely dissolved.

## Preparation of oleylammoium iodide solution for halide exchange

0.1844 g of $PbI_2$ (0.4 mmol), hexane (10 mL), oleylamine (0.4 mL) and oleic acid (0.4 mL) were combined in a 20 mL vial. The resulting mixture was heated at 50 °C under continuous stirring overnight to completely dissolve the salt.

## Synthesis of CsPbBr₃, CsPbI₃

In a typical synthesis, 6 mL of the corresponding $PbX_2$ precursor solution in a 20 mL glass vial was heated on a hot-plate until the temperature of the precursor solution reached 175 °C (for $PbBr_2$) or 150 °C (for $PbI_2$). Afterwards, 400 μL of the Cs-OL stock solution (pre-heated at 120 °C) was swiftly injected into the lead-halide salt solution under vigorous stirring (1000 rpm). After 5 s, the vial was cooled in an ice-water bath to quench the reaction. Subsequently, the colloidal dispersion obtained was purified by centrifugation at 8000 rpm (7311 g) for 10 min. The supernatant was then discarded to remove the unreacted precursors and ligands, and the precipitate was redispersed in 5 mL of toluene. The colloidal dispersion was centrifuged again at 5000 rpm (2856 g) for 8 min to remove the largest particles in the sediment.

## Synthesis of CsPb(Br,I)₃ perovskite NCs

Firstly, lead iodide solutions were prepared by dissolving 0.2 mmol of $PbI_2$ powder in a mixture of 10 mL toluene, 0.2 mL oleylamine and 0.2 mL oleic acid at 100 °C under continuous stirring until the precursor dissolved. Then, the appropriate amount of $PbI_2$-ligand solution was added to the parent $CsPbBr_3$ NC colloidal solution to initiate halide ion exchange. The target Br:I ratio was 3:7. The reaction mixtures were stirred at 800 rpm for 30–40 min at 40 °C.

## Purification process for preparing NCs with different defect densities

1 mL of synthesised NCs solution (denoted as pristine) was added into a 50 ml centrifuge tube, mixing with 0.5 mL of methyl acetate. The mixture was then centrifuged at 8000 rpm (7311 g) for 10 min. The precipitates were collected and redispersed in 1 mL hexane (donated as 1st wash). This same process was repeated to prepare the most defective NCs samples (denoted as 2nd wash). The 0th, 1st, and 2nd samples were then used for compositional and optical characterisation.

## Characterization

### Ultraviolet-Visible Absorption (UV-Vis).

Spectra were recorded on a Shimadzu UV-VIS-NIR Spectrophotometer, UV-3600Plus. The absorption spectra were measured with NC solutions in quartz cuvette, the baseline of the instrument is calibrated with pure hexane in quartz cuvette.

### Photoluminescence (PL) and photoluminescence quantum yield (PLQY).

These measurements were performed using a UV-NIR absolute PLQY spectrometer (HAMAMATSU, Quantaurus-QY plus; excitation wavelengths of 375 nm to 575 nm). The optical density of the pristine (low defect sample) and doubly purified (high defect sample)

samples were tuned to be less than 0.2 with hexane dilution to minimise the concentration effect of photon recycling and inter-NC energy transfer[28,43].

### X-ray photoemission spectroscopy.

XPS data was acquired using a Kratos Axis SUPRA using monochromated Al $K_\alpha$ (1486.69 eV) X-rays at 12 mA emission and 15 kV HT (180 W), with an analysis area of 700 × 300 μm². All data was recorded at a base pressure of below $9 \times 10^{-9}$ Torr and at room temperature (294 K). The results were analysed using the software CasaXPS v2.3.19PR1.0.

### Pump-Probe Transient Absorption Spectroscopy (PP-TA).

The output of a Ti:sapphire amplifier system (Spectra Physics Solstice Ace) operating at 1 kHz and generating ~100 fs pulses (fundamental, 800 nm) was split into pump and probe pulses. The 400 nm pump pulses were created by sending the 800 nm fundamental beam of the Solstice Ace through a second harmonic generating beta barium borate (BBO) crystal (Eksma Optics). The pump is blocked by a chopper wheel rotating at 500 Hz. The broadband probe (330–700 nm) is generated by focusing the 800 nm fundamental beam onto a $CaF_2$ crystal (Eksma Optics, 5 mm) connected to a digital motion controller (Mercury C-863 DC Motor Controller). The pump-probe delay (100 fs to 2 ns) was controlled by a mechanical delay stage (Thorlabs DDS300-E/M). The transmitted pulses were collected with a monochrome line scan camera (JAI SW-4000M-PMCL, spectrograph: Andor Shamrock SR-163). The spectrum was taken with liquid samples inside a 1 mm thick cuvette. The concentrations of the samples were adjusted to an absorbance of 1 OD at 400 nm in hexane. The carrier density was calculated using the method reported in ref. 48. Based on the Poisson distribution of the initial photon occupancies in NCs, the probability of one NC containing $i$ number of excitons is given by $p_i = \left( \frac{\langle N^i \rangle}{i!} \right) e^{(-\langle N \rangle)}$, where $N$ is the average number of excitons per nanocrystals, $\langle N \rangle = \sigma j$, $p_i$ the probability of a nanocrystal containing $i$ excitons, $j$ is the pump fluence, and $\sigma$ is the absorption cross-section of the nanocrystals. TA signal intensity is directly proportional to $\sigma j$ when there is less than 1 exciton per NCs, i.e., $I_{TA} \propto 1 - e^{(-\langle N \rangle)} = 1 - e^{(-\sigma j)}$. The estimated $\sigma$ for Br-based NCs is $3.1 \times 10^{-15}$ cm², for mixed Br/I NCs is $4.2 \times 10^{-15}$ cm², and for I-based NCs is $6.2 \times 10^{-15}$ cm². The carrier density, $n_0$ is estimated based on the volume of the NCs obtained by TEM, $n_0 = \frac{\langle N \rangle}{V_{NC}}$.

### Pump-Push-Probe Transient Absorption Spectroscopy (PPP-TA).

The PPP-TA experiments are based on a modification of a commercial transient absorption spectroscopy setup (Helios, Ultrafast Systems). The fundamental 800 nm (1 kHz, <100 fs pulse duration) output, generated by a Ti:sapphire regenerative amplifier (Solstice, Spectra Physics, Newport Corp.) was divided into three portions to derive pump, push and probe pulses. The first portion is directed into a β-barium borate (BBO) crystal to generate second-harmonic output (400 nm) which acts as the pump. The second portion of the 800 nm is directed towards an optical parametric amplifier (TOPAS Prime, Spectra-Physics) and a frequency mixer (Niruvis, Light Conversion) to generate the NIR push pulse (1300 nm). The remaining 800 nm pulses are focused into a sapphire or a YAG crystal to generate broadband visible probe (450–750 nm) and NIR probe (900–1400 nm), respectively.

In the present setup, the push arrives 10 ps later than the pump by controlling the relative optical path lengths, while the probe beams are delayed w.r.t. the pump by automated mechanical delay stages. The three beams are focused onto the sample surface at the same spot (diameter ~0.5 mm), and the transmitted probe is collected by a CCD spectrometer. To mitigate shot-to-shot noise, we use another CCD spectrometer to measure fluctuations in a reference beam that is split off from the probe before it hits the sample. A mechanical chopper is in place in the pump beam path to modulate

at 500 Hz, blocking every other pulse and synchronised with the CCD to relay the TA signal.

Under the same experimental conditions, we record the pump-probe signal with the push beam either blocked or open to obtain PP and PPP spectral/dynamical features, respectively.

## Density Functional Theory (DFT)

Structural relaxations and electronic properties calculations were performed based on density-functional theory as implemented in the Vienna Ab initio Simulation Package (VASP)[57]. The exchange and correlation functional were described using the Perdew–Burke–Ernzerhof generalised gradient approximation (GGA)[58]. The all-electron projector augmented wave (PAW) method was adopted to describe the electron-ion interactions[59], treating $5s^2 5p^6 6s^1$, $6s^2 6p^2$, $4s^2 4p^5$ and $5s^2 5p^5$ as valence electrons for Cs, Pb, Br and I atoms, respectively. To ensure that all enthalpy calculations were well converged to the level of 1 meV per atom, the plane-wave energy cut-off was set at 500 eV, and Monkhorst-Pack k-point grids with a reciprocal space resolution of $2\pi \times 0.03$ Å$^{-1}$ in the Brillouin zone were chosen[60]. Structural relaxations were performed with forces converged to less than 0.001 eV Å$^{-1}$. The defect structure of $CsPbX_3$ was constructed by removing a halogen atom from the 2×1×1 supercell. The atomic coordinates of the optimized computational models, calculated density of states and band structures can be found in Supplementary Data 1.

## Hot carrier cooling dynamics modelling

The midpoint method was employed as a numerical solution to the coupled differential Supplementary Eq. S6-9 in the Supplementary Information. The kinetics were simulated for a range of $\tilde{I}$ (push fluence) and $N_t$ (total defect density), and a Gaussian-convolved exponential decay time extracted ($\tau_{hot}$) for each. The fit parameters are presented in Supplementary Table 5.

## Data availability

The raw data for this paper and Supplementary Information have been deposited to the Oxford Research Data Archive, with the (https://doi.org/10.5287/ora-nrxxqxzz4). The optimized atomic coordinates of the structures used for DFT calculations, as well as the electronic structure and band structure calculations generated in this study are provided in Supplementary Data 1.

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

## Acknowledgements

J.Y. and R.L.Z.H. acknowledge support from a UK Research and Innovation (UKRI) Frontier Grant (no. EP/X029900/1), awarded via the European Research Council Starting Grant 2021 scheme. J.Y. also gives thanks to Cambridge Philosophical Society for the Research Studentship Grant and Churchill College for various travel and research grants. N.M. and A.A.B. acknowledge support from the European Commission through the Marie Skłodowska-Curie Actions Project (PeroVIB, H2020-MSCA-IF-2020-101018002). Y.W.Z. acknowledges funding from the National Natural Science Foundation of China under Grant No. 12304036, the Open Project of Guangdong Provincial Key Laboratory of Magnetoelectric Physics and Devices (No. 2022B1212010008), the Guangdong Basic and Applied Basic Research Foundation (2023A1515010071), the Guangzhou Basic and Applied Basic Research Foundation (SL2022A04J00048), and the Fundamental Research Funds for the Central Universities, Sun Yat-sen University (23xkjc016). S.D.S., L.D. and J.M. acknowledge funding support from the European Research Council under the European Union's Horizon 2020 research and innovation programme (HYPERION, 756962; PEROVSCI, 957513). J.M. acknowledges funding from a Marie Sklodowska-Curie Postdoctoral Fellowship via UKRI Horizon Europe Guarantee (grant number EP/X025764/1). S.D.S. thanks the Royal Society and Tata Group (UF150033). L.D. thanks the Cambridge Trusts and the China Scholarship Council for funding. J.M. acknowledges funding from Marie Sklodowska-Curie Postdoctoral Fellowships via UKRI Horizon Europe Guarantee (no. EP/X025764/1) and the National Natural Science Foundation of China (no. 62204049). L.v.T. thanks the Winton Programme for the Physics of Sustainability and the Engineering and Physical Sciences Research Council for funding. A.A.B. acknowledges support from the Royal Society and Leverhulme Trust. L.P. acknowledges support from the Spanish Ministerio de Ciencia e Innovación through Ramón y Cajal grant (grant no. RYC2018-026103-I), the Spanish State Research Agency (grant nos. PID2020-117371RA-I00 and TED2021-131628A-I00), and a grant from the Xunta de Galicia (grant no. ED431F2021/05). Y.-T.H. and R.L.Z.H. thank the Engineering and Physical Sciences Research Council (EPSRC, grant no. EP/V014498/2). R.L.Z.H. thanks the Royal Academy of Engineering through the Research Fellowships scheme (no. RF\201718\17101), as well as the Centre of Advanced Materials for Integrated Energy Systems (CAM-IES; EPSRC grant no. EP/T012218/1). The authors thank Dr. Mark Isaacs for XPS measurements. The author also would like to acknowledge that the X-ray photoelectron (XPS) data was acquired at the EPSRC National Facility for XPS ("HarwellXPS", EP/Y023587/1, EP/Y023609/1, EP/Y023536/1, EP/Y023552/1 and EP/Y023544/1; these are associated with Dr. Mark Isaacs).

## Author contributions

J.Y. and R.L.Z.H. conceived the project. J.Y., L.D. and C.O.-M. synthesized and purified the nanocrystals with help from L.P., J.Y. conducted pump-probe transient absorption measurement with help from L.D., N.C.G. and P.G., N.M. conducted the pump-push-probe transient absorption measurements for nanocrystals with different compositions and defect densities and analysed the data, with input from Z.M.C. B.P.C. performed the detailed kinetic modelling under the supervision of A.B., Z.Q.C. did the DFT calculations under the supervision of Y.W.Z., J.Y. and X.-B.F. performed energy-dependent PLQY measurement. J.M. helped with TCSPC measurements under the supervision of S.D.S. L.v.T. and Z.Y. performed the TEM measurements. Y.-T.H. contributed to the TA analysis. A.R. and R.L.Z.H. supervised the work. All authors contributed to writing and editing the manuscript.

## Competing interests

The authors declare no competing interests.
