## [Peer Review File · Nature Communications]

Extending the Defect Tolerance of Halide Perovskite Nanocrystals to Hot Carrier Cooling DynamicsReviewer #1 (Remarks to the Author):

In this work, Ye et al. investigated the effect of defect tolerance of perovskite nanocrystals on hot carriers cooling process. They found that HCs are directly captured by traps and deeper traps cause faster HC cooling. The results are very useful contribution for developing highly efficient HCs solar cells. I recommend this paper for publication after revision on the following point:

1. On page 3, line 53, some appropriate references should be added for this statement 'HCs release this excess energy through ultrafast (<100 fs) carrier-carrier scattering, followed by carrier-phonon interaction events (<1 ps) to reach the lattice temperature'
2. In Figure 1g, perovskite NCs under low and high defects show almost same PLQY, the author can explain this phenomenon.
3. In this work, the author employed the number of purification to prepare perovskite NCs with different defect densities, which is confirmed by PLQY and XPS measurements. But for the defect densities these evidences are not sufficient. The defect densities of the perovskite NCs can be quantitatively studied through the space-charge-limited-current technique.
4. The authors present the perovskite defect tolerance to HC cooling dynamics in this work, but hardly put in relation to each other. For example, it remains unclear to me what the main physical process of deeper traps causing faster HC cooling is and how traps will have a significant influence on HC cooling process. Generally speaking, the discussion is so speculative in places.
5. Some poor statements including many grammatical errors were seen in the manuscript. They should be extensively revised.
6. Reason for the author to choose CsPbX₃ NCs?
7. There is an obvious shift for GSB peak in the Figure 2a~c. What caused this change? The author should provide discussion about this.
8. Figure S7h~g are blurry, please update them. Moreover, perovskite NCs show high initial HC temperature (T_c) excited under high excitation fluence. Why is the T_c of the sample (77.72 uJ cm⁻²) higher than that of the one (116.58 uJ cm⁻²), as shown in Figure S7h~g.

Reviewer #2 (Remarks to the Author):

Manuscript: NCOMMS-24-17713

Title: Extending the Defect Tolerance of Halide Perovskite Nanocrystals to Hot Carrier Cooling Dynamics

The authors of the manuscript "Extending the Defect Tolerance of Halide Perovskite Nanocrystals to Hot Carrier Cooling Dynamics" investigate how traps affect hot-carrier relaxation in lead-halide perovskite materials. Using ultrafast transient absorption spectroscopy and photoluminescence measurements, they show that the hot carriers are directly captured by traps, instead of going through an intermediate cold carrier, addressing current important disunion within the community. The authors believe that these results will be important for future design of materials for hot-carrier solar cells. And I agree with them.

The manuscript is exceptionally well written with nice and clear figures illustrating the results of the work. I'm convinced that the conclusion, that the hot carriers are dependent on trap depth,

is well proven and supported by data. With that, I would recommend the manuscript to be published in Nature Communications with minor revisions.

Prior to the possible publication, I would like the authors to address the following issues:

- 1) All the TA figures in the manuscript are plotted in wavelengths (nm) and “ $\Delta T/T$ ” while in the SI most of the TA data are plotted in energy scale (eV) and as “ $-\Delta T/T$ ” and only the last Supplementary Fig.10 is again in wavelength. While I do appreciate using different scales for different purposes, in this case it is really confusing, for someone trying to understand the observed dynamics. Is there any specific reason for presenting the same datasets in such a different way? If not, these should be unified, to avoid further confusion.
- 2) Following the TA spectral evolution for better understanding the kinetics in Fig.2 (c, d, e) is quite difficult already, as mentioned above, however for the panel e (kinetics of CsPbI₃) it is rather impossible. The kinetics are plotted at 750-760 nm, but the corresponding TA data sets (both the matrix [a, b c] and differential spectra [d, e, f]) in the SI are shown only up to ~710 nm (1.75 eV). Thus, following the spectral evolution at that wavelength is, for the reader, impossible. The data should be shown.
- 3) Throughout the manuscript, when describing the measured lifetimes, the authors write the values and their errors down to two decimals (for example 2.63±0.41 ps). This, in fact means, that the lifetimes were determined with a precision down to tens of femtoseconds. However, I highly doubt this! In the Methods section, the laser used for TA is described as generating ~100 fs pulses, which together with ultrafast artifacts like GVD (chirp) and cross-phase modulation typically generates response function of at least ~150 fs, unless special precautions are made. This is even more striking in the SI, where the Fig.11 even shows lifetimes determined down to third decimal, thus with a precision down to a single femtosecond! I’m convinced, these numbers are just an artifact of the fitting procedure and that they have no physical (nor chemical) meaning as the data (and even the method) simply does not allow such a precision. These numbers should be rounded throughout the manuscript to a single decimal; hundreds of femtoseconds is a reasonable precision in this configuration, I’m convinced.
- 4) I really like the utilization of the 3 pulse TA in this case, the message it delivers is solid! The simple kinetic model can also quite surprisingly well describe the measured data. Only in one case the fit differs quite significantly from the data, which is rather surprising. Could the authors comment on why they think the measured Tau(hot) for CsPb(Br/I)₃ are lower than the model would predict?
- 5) The Supplementary Fig. 1 has two panels “e”.

Reviewer #3 (Remarks to the Author):

The manuscript by Hoyer and co-workers reports a study of hot-carrier cooling dynamics in CsPbBr₃, CsPbI₃, and CsPb(Br/I)₃ NCs with femtosecond pump-probe and pump-push-probe transient absorption spectroscopy. The NCs were washed to deliberately create traps to investigate the effect of traps on the hot carrier cooling process in the three systems. The authors find that the decay process follows the defect-tolerance nature of the NCs, like that of band edge carrier decay. They found that the hot carriers can be directly captured by traps before going through intermediate cold carriers, and thus deeper traps lead to faster cooling. The kinetic modeling used to explain the hot carrier cooling lifetimes fits well with the

experimental data. The experiments are highly innovative, and the results are very interesting. Therefore, the manuscript can be interesting for publication in Nature Communications. I recommend to publish it. But a few issues need the author's attention.

1. Could the author explain briefly in the text how they calculate the carrier density shown in Figure 4?
2. What is the fitting window to obtain the temperature vs time graph shown in Supplementary Fig. 7-9? Could the author justify their choice of selecting the fitting windows?
3. In Fig.3j, the maximum carrier temperatures for samples with different defects are different, the low defects sample can reach almost 4000K, while the high defect sample only 1000K, could the author explain?
4. Why do the CsPbI₃ nanocrystals degrade after two washing cycles?
5. Is the model used to fit the PPP cooling data also capable of describing the PP cooling data?
6. Why the authors didn't compare the results with CsPbCl₃ NCs, which are less defect-tolerant?
7. Did the authors try to passivate and then see if the decay dynamics recover to the original?
8. Authors might consider referring to relevant articles like Nano Lett. 2017, 17, 9, 5402–5407; J. Phys. Chem. Lett. 2021, 12, 36, 8732–8739.

Reviewer 1:

In this work, Ye et al. investigated the effect of defect tolerance of perovskite nanocrystals on hot carriers cooling process. They found that HCs are directly captured by traps and deeper traps cause faster HC cooling. The results are very useful contribution for developing highly efficient HCs solar cells. I recommend this paper for publication after revision on the following point:

We thank the reviewer for their time in evaluating our paper, and are delighted to see their strongly positive appraisal, particularly noting the importance of our results to the wider community. We have now addressed all points raised, as detailed below.

1. On page 3, line 53, some appropriate references should be added for this statement 'HCs release this excess energy through ultrafast (<100 fs) carrier-carrier scattering, followed by carrier-phonon interaction events (<1 ps) to reach the lattice temperature'

Response 1.1: We have now added references to each process indicated on page 3:

Carrier-carrier scattering:

- [12] Schoenlein, R. W., Lin, W. Z., Ippen, E. P. & Fujimoto, J. G. Femtosecond hot - carrier energy relaxation in GaAs. *Applied Physics Letters* **51**, 1442-1444 (1987). <https://doi.org/10.1063/1.98651>
- [13] Richter, J. M. *et al.* Ultrafast carrier thermalization in lead iodide perovskite probed with two-dimensional electronic spectroscopy. *Nature Communications* **8**, 376 (2017). <https://doi.org/10.1038/s41467-017-00546-z>
- [14] Karaman, C. O., Bykov, A. Y., Kiani, F., Tagliabue, G. & Zayats, A. V. Ultrafast hot-carrier dynamics in ultrathin monocrystalline gold. *Nature Communications* **15**, 703 (2024). <https://doi.org/10.1038/s41467-024-44769-3>

Carrier-phonon interactions:

- [10] Fu, J. *et al.* Hot carrier cooling mechanisms in halide perovskites. *Nature Communications* **8**, 1300 (2017). <https://doi.org/10.1038/s41467-017-01360-3>
- [15] Dai, L., Ye, J. & Greenham, N. C. Thermalization and relaxation mediated by phonon management in tin-lead perovskites. *Light: Science & Applications* **12**, 208 (2023). <https://doi.org/10.1038/s41377-023-01236-w>
- [16] Sourabh, S. *et al.* Hot carrier redistribution, electron-phonon interaction, and their role in carrier relaxation in thin film metal-halide perovskites. *Physical Review Materials* **5**, 095402 (2021). <https://doi.org/10.1103/PhysRevMaterials.5.095402>
- [17] Mondal, N., Carwithen, B. P. & Bakulin, A. A. Alloying metal cations in perovskite nanocrystals is a new route to controlling hot carrier cooling. *Light: Science & Applications* **12**, 276 (2023). <https://doi.org/10.1038/s41377-023-01316-x>
- [18] Hopper, T. R. *et al.* Hot Carrier Dynamics in Perovskite Nanocrystal Solids: Role of the Cold Carriers, Nanoconfinement, and the Surface. *Nano Letters* **20**, 2271-2278 (2020). <https://doi.org/10.1021/acs.nanolett.9b04491>

"In general, HCs release this excess energy through ultrafast (<100 fs) carrier-carrier scattering¹²⁻¹⁴, followed by carrier-phonon interaction events (<1 ps) to reach the lattice temperature^{10,15-18}."

2. In Figure 1g, perovskite NCs under low and high defects show almost same PLQY, the author can explain this phenomenon.

Response 1.2: There is only a small change in the PLQY of these I-based nanocrystals with defect density because the traps formed are predominantly shallow in this case. These iodide vacancies have a small barrier to electron capture, but a significantly larger barrier to hole capture, thus leading to a low likelihood of electron-hole non-radiative recombination. These traps are therefore comparatively benign, and our results are fully consistent with observations made recently by us¹ and the Alivisatos group², and are the basis for defect tolerance in halide perovskites to cold carriers^{3,4}.

We have added the following to the main text (Page 6):

“Indeed, we observe only a small change in PLQY with increasing excess energy in CsPbI₃ NCs, and this was not exacerbated by introducing a higher defect density (Fig. 1g), due to the shallow nature of the traps and the defect tolerance of I-based NCs, which is consistent with previous literature^{43,45}.”

3. In this work, the author employed the number of purification to prepare perovskite NCs with different defect densities, which is confirmed by PLQY and XPS measurements. But for the defect densities these evidences are not sufficient. The defect densities of the perovskite NCs can be quantitatively studied through the space-charge-limited-current technique.

Response 1.3: We are grateful for the opportunity to revisit this point, and thank the reviewer for the suggestion to consider SCLC measurements. To answer this question, in this response, we firstly discussed the challenges of using SCLC to accurately quantify the defect density of perovskite NCs before going over the extra measurements that we made instead by photothermal deflection spectroscopy (PDS) to fully address this comment from the reviewer.

SCLC measurements are indeed popular in the literature for measuring defect density, but there are many instances where it is not used properly. It is very common to find reports where plots of current vs. voltage do not have all three of the 1) initial ohmic regime, 2) steep trap filling regime, and 3) Child's regime. Often, devices would exhibit a steep increase in current but not enter into a quadratic Child's regime. Furthermore, the initial regime may not be strictly linear due to a non-ideal single-carrier device being made, or because of a distribution of traps, not only in the bulk, but also at the interfaces of the test device. We delved deeper into recent reports of SCLC measurements made of halide perovskites (Table R1, below), and found the trap densities obtained to vary over a wide range, from 10^{13} cm^{-3} to 10^{18} cm^{-3} . Whilst some of this variation would come from actual differences in trap density, we suspect that an important source of this variation comes from non-idealities in the SCLC measurement, or how the single carrier devices were made. For example, in Table R1, it can be seen that two reports of SCLC of CsPbI₃ NCs reported defect densities of 10^{14} cm^{-3} in one case, and 10^{18} cm^{-3} in the other⁵⁻⁷. Similarly, for CsPbBr₃, even for the same CsPbBr₃ NC films, the defect density reported in the literature show three order of magnitude differences^{8,9}, ranging from 10^{15} to 10^{18} cm^{-3} . An important source of error in these SCLC measurements is the effect of the ligands used. CsPbX₃ NCs cannot easily have their long-chain organic ligands exchanged for short-chain ligands in films, and these long-chain ligands will affect the current densities obtained in diodes. It is therefore difficult to extract the accurate trap density within the inorganic NC core itself by SCLC.

Table R1 | Defect densities obtained by space charge limited current method.

Materials	Defect Densities (cm ⁻³)	Ref.
MAPbBr ₃ single crystal	1.3 × 10 ¹³	10
MAPbBr ₃ single crystal	4.18 × 10 ¹⁰	11
CsPbBr ₃ single crystal	8.81 × 10 ⁹	12
(FA _{0.33} Cs _{0.67} Pb(I _{0.7} Br _{0.3}) ₃) Bulk film	2.52 × 10 ¹⁶ (hole)	13
(FA _{0.33} Cs _{0.67} Pb(I _{0.7} Br _{0.3}) ₃) Bulk film	6.72 × 10 ¹⁶ (electron)	13
Cs _{0.03} (FA _{0.90} MA _{0.10}) _{0.97} Pb(I _{1-x} Br _x) ₃ Bulk Film	7.01 × 10 ¹⁵	14
CsPbBr ₃ Nanocrystal Films	8.82 × 10 ¹⁵	8
CsPbBr ₃ Nanocrystal Films	2.04 × 10 ¹⁶	15
CsPbBr ₃ Quantum Dots Films	3.00 × 10 ¹⁸	9
CsPbI ₃ Nanocrystal Films	1.26 × 10 ¹⁷	5
Cs(PbZn)I ₃ Nanocrystal Films	1.75 × 10 ¹⁶	5
CsPbI ₃ Nanocrystal Films	8.30 × 10 ¹⁴	6
CsPbI ₃ Nanocrystal Films	1.89 × 10 ¹⁸	7
CsPbCl ₃ Nanocrystal Films	4.55 × 10 ¹⁷	16
CsPbClBr ₂ Nanocrystal Films	2.16 × 10 ¹⁸	17
Cs(PbCu)ClBr ₂ Nanocrystal Films	1.87 × 10 ¹⁷	17

Therefore, we understand what the reviewer is trying to get at, but we believe that SCLC is not the best way to confirm the changes in defect density across the samples. PDS, on the other hand, is a simpler method to qualitatively compare changes in defect density through the sub-bandgap absorbance and Urbach energy. We collected these new data and added them to Supplementary Fig. 2 (copied below). Herein, we show that for all compositions, there is an increase in the sub-bandgap absorption with an increase in the number of purification steps, confirming an increase in sub-bandgap defect density. Consistent with this, we observed an increase in the Urbach energy as the number of purification steps increased, implying increased disorder as the ligands and surface halides are increasingly removed. This is in agreement with our previous study¹⁸, as well as the XPS measurements made in the present work, which show a removal of surface halides with an increasing number of purification steps (Supplementary Fig. 2a-c)

In short, we believe that our PLQY, PDS and XPS results are fully consistent with each other, and clearly show an increase in defect density in the inorganic NC core upon an increase in the number of purification steps.

We have added the following to the main text on Page 5:

“The increase in defect density was further confirmed by photothermal deflection spectroscopy (PDS), which show an increase in the sub-bandgap absorption and Urbach energy as the NCs were washed with polar antisolvents an increasing number of times (Supplementary Fig. 2d-f). By contrast, there was very little change in the PLQY and PL lifetime of the narrow-gap CsPbI₃ (Fig. 1a, Supplementary Fig. 4c), despite substantial changes in surface composition (Supplementary Fig. 2c) and sub-bandgap absorbance (Supplementary Fig. 2f) after purification. This is consistent with the greater defect tolerance of CsPbI₃, due to the dominant iodide vacancies being shallow.”

The modified supplementary Supplementary Fig. 2 is copied below for convenience:

Supplementary Fig. 1 | Compositional analysis based on X-ray photoemission spectroscopy. a. Relative Pb/Br ratio for CsPbBr₃ NCs after washing. **b.** Relative Pb/Br and I ratio for CsPbBr_xI_{3-x} NCs after washing. **c.** Relative Pb/I ratio for CsPbI₃ NCs after washing. **d-e.** Photothermal deflection spectroscopy (PDS) measurement for CsPbX₃ NCs with different defect densities and their Urbach energy. The PDS and Urbach energy fitting are measured and calculated following the procedure reported in our previous work¹.

4. The authors present the perovskite defect tolerance to HC cooling dynamics in this work, but hardly put in relation to each other. For example, it remains unclear to me what the main physical process of deeper traps causing faster HC cooling is and how traps will have a significant influence on HC cooling process. Generally speaking, the discussion is so speculative in places.

Response 1.4: We understand the reviewer's concern. We have modified the configuration coordinate diagram in Supplementary Fig. 5 (copied below) to better illustrate our model for how traps and their energetic position relative to the band-edge affect hot carrier cooling. These trap states introduce an extra relaxation channel for the hot carriers, and Marcus theory predicts that the transfer rate between the excited states to the trap states (k_t) is related to the energy offset between these two states (ΔE), as well as the degree of coupling (H) between the trap states and carriers, as discussed in Supplementary Note 1 (Eq. S2, copied below). When the defects are shallow, ΔE is negligible, and the transfer rate would be similar regardless of the number of defects present in the samples. This can be found in Supplementary Table 1, where the fitted constant, b , which is related to transfer rate for shallow traps (k_{ts}), is very close for low defect density CsPbI₃ NCs (0.09) and high defect density CsPbI₃ NCs (0.11). However, when the trap is deep, ΔE is no longer negligible, and there is more limited ability for the trapped carriers to return to the bandedge as free carriers. As a result, the differences between the fitted constant, b , for low and high defect density Br-based and Br/I-mixed NCs is much larger than in I-based NCs, as shown in Supplementary Table 1. This implies that there will be more direct hot carrier trapping when the defect is deep, due to the higher transfer rate to defect states compared to when the defect is shallow, and this can be visualised by the excess area (ΔA_1) shown in Supplementary Fig. 5. This additional hot carrier trapping speeds up the whole cooling process, and hence reduces the time taken for hot carrier cooling.

$$k_t(\delta) = \frac{2\pi}{\hbar} |H|^2 \frac{1}{\sqrt{4\pi\lambda k_B T_L}} e^{-\frac{(\lambda+\epsilon+\Delta G^0)^2}{4\lambda k_B T_L}} \quad (\text{S2})$$

Supplementary Fig. 5 | Schematic energy diagram describing the interaction between carriers and shallow or deep traps in PNCs.

We have modified the main text in page 6:

“The apparent ‘tolerance’ of HCs to traps in CsPbI₃ could be due to these traps being shallower than in CsPbBr₃ and CsPbBr_xI_{3-x} NCs, such that the overlap between the conduction band and trap state potential energy surfaces is smaller with a smaller energy offset for the HCs to be trapped¹⁹.”

We have modified the supplementary information (Supplementary Note 1):

$$-\frac{dn}{dt} = k_1 n + k_2 n^2 + k_3 n^3 = k_2 n^2 + \int_0^\infty k_t(\epsilon) n(\epsilon, t) d\epsilon + e^{\frac{\Delta E}{k_B T}} \int_0^\infty k_t(\epsilon) n(\epsilon, t) d\epsilon \quad (\text{S1})$$

where n is carrier density, k_1 the trap-assisted non-radiative recombination rate constant, k_2 the bimolecular radiative recombination rate constant, and k_3 the Auger recombination rate constant, which is negligible in this case since we measured the PLQY at relatively low excitation power. The modified model implies that the k_1 rate constant is governed by the Marcus theory of charge transfer from free carriers to the potential energy surface of trap states, and correlated the trap transfer rate, k_t with the excess energy that the excited carriers have (δ).

We then arrive at Eq. S2.

$$k_t(\delta) = \frac{2\pi}{\hbar} |H|^2 \frac{1}{\sqrt{4\pi\lambda k_B T_L}} e^{-\frac{(\lambda+\epsilon+\Delta E)^2}{4\lambda k_B T_L}} \quad (\text{S2})$$

5. Some poor statements including many grammatical errors were seen in the manuscript. They should be extensively revised.

Response 1.5: We apologize for these errors, and have now thoroughly gone through the entire paper and supplementary information to correct all grammatical errors and typos to ensure that the wording used is precise.

For example: Page 2

“internationally introduced traps” was corrected to “*intentionally introduced traps*”

6. Reason for the author to choose CsPbX₃ NCs?

Response 1.6: We chose the CsPbX₃ NC system for two reasons:

Firstly, from an applications point of view, CsPbX₃ NCs are considered promising candidates for ultra-high-definition displays, due to their narrow emission linewidth, high PLQYs, and finely tuneable wavelength through the size and composition. CsPbX₃ show better thermal stability than their hybrid counterparts, especially methylammonium-based NCs²⁰. Also, there are minimal NC-ligand interactions when the A-site cation is Cs⁺. This leads to easier surface ligand removal compared to hybrid perovskite NCs, which helps with preparing NCs with different surface defect densities.

Secondly, we need to have a system where we can monitor the defect density easily and rapidly. Bulk perovskite thin films usually have much lower PLQY, due to grain boundary defects and non-passivated surfaces. Preparing and probing the defect densities in bulk systems is not straightforward when we need samples with controllable defect densities. In contrast, CsPbX₃ NCs can achieve near unity PLQY, and the relative defect densities can be easily and quickly probed by fast PLQY measurements. We hope that our work will inspire the wider community to investigate the role of traps on hot carrier cooling in broader classes of halide perovskites, and the experimental protocols we have developed in this work will pave the way for the wider community.

We modified the main text on Page 4:

“To elucidate the role of traps on HC cooling lifetime, CsPbX₃ NCs were selected as the material system of interest rather than LHP thin films. This is because of the capability of introducing different defect densities intentionally by controlling the NC surface chemistry through multiple antisolvent purification steps¹⁸ and also its wide applications in display, optical communication and energy harvesting²¹⁻²⁴. This enables a more direct correlation between the effects of defects and non-radiative recombination, which affects the photoluminescence quantum yield (PLQY).”

7. There is an obvious shift for GSB peak in the Figure 2a~c. What caused this change? The author should provide discussion about this.

Response 1.7: We thank the reviewer for the suggestion. The early-time red-shift in GSB shown in Fig. 2a and b is an indication of hot carrier cooling. The initial broadband GSB indicates the carrier distribution to have energy in excess of the bandgap, *i.e.*, so-called hot carriers. The gradual red-shift and narrowing of the GSB peak illustrates the loss of hot carrier energy when they relax to the band-edge, where the GSB peak position matches the ground state absorption peak. This hot carrier cooling behaviour can also be monitored by the negative PIA signal due to bandgap renormalization. This shift is more obvious when we plot the GSB spectra at smaller time intervals, as shown in Supplementary Fig. 8-10 panel d-f.

Following the suggestion, we modified the main text on Pages 6 and 7:

“Fig. 2a and b show the TA spectra (within 0.5-30 ps, note labelling of y-axes as $\Delta T/T$) of the low and high defect density CsPbBr₃ samples. Three main spectral signatures are often observed in perovskite TA spectra: i) a high-energy, broad negative photo-induced absorption (PIA), usually caused by a refractive index change, but can also be ascribed to polaronic signatures⁴⁸, ii) positive ground state bleach (GSB), attributed to the depopulation of band-edge carriers, and iii) a negative sub-bandgap short-lived PIA, which is often ascribed to band-gap renormalization (BGR). The early red-shift of the GSB (feature ii) is an indication of hot carriers with excess energy relaxing to the band-edge (steady-state absorption onset). The decay of this sub-gap PIA is correlated with the rise of the GSB (due to state-filling during HC cooling). As we have intentionally introduced defects into the NCs, the sub-bandgap region also has a positive trap bleach (TB) signature overlaid with the negative BGR feature. Fig. 2c displays the sub-bandgap kinetics (at 535-545 nm) which show a clear difference between the low (red dots) and high (blue dots) defect density CsPbBr₃ NCs. While initially negative due to BGR upon excitation, $\Delta T/T$ then evolves into a positive TB feature within 1 ps, indicating carriers are being trapped into the defect states even during the HC cooling time window. The TB subsequently decays over ~100 ps. Similarly, the sub-gap kinetics are also dependent on defect density in the mixed-halide NCs (Fig. 2d). However, no obvious change is observed for pure I-based NCs (Fig. 2e).”

8. Figure S7h~g are blurry, please update them. Moreover, perovskite NCs show high initial HC temperature (T_c) excited under high excitation fluence. Why is the T_c of the sample (77.72 $\mu\text{J cm}^{-2}$) higher than that of the one (116.58 $\mu\text{J cm}^{-2}$), as shown in Figure S7h~g.

Response 1.8: Thanks for pointing this out. We have now improved the figure resolution and the font size to be consistent with the other figures in the paper. Also, we have looked into our fitting and calculations for the Br-based perovskite NC series. The previous higher value for T_c is due to an outlier in the TA signal, so that it appears to be at higher temperature after normalization to the steady lattice temperature. This has now been corrected by removing the outlier for normalization, as shown in Supplementary Fig. 8 (copied below).

Supplementary Fig. 8 | Hot carrier cooling kinetics, based on pump-probe transient absorption spectroscopy for CsPbBr₃ NCs. **a-c**, TA maps, **d-f**, TA spectra and **h-g**, extracted charge-carrier temperatures vs. pump-probe delay of **a, d, h**, pristine (low defect density), **b, e, i**, single-purified (moderate defect density), and **c, f, g**, doubly-purified (high defect density) CsPbBr₃ perovskite nanocrystal solutions. The pump wavelength was 400 nm, repetition rate 1000 Hz, and the maps and spectra were recorded under a fluence of 194.32 $\mu\text{J cm}^{-2}$.

Reviewer 2:

The authors of the manuscript “Extending the Defect Tolerance of Halide Perovskite Nanocrystals to Hot Carrier Cooling Dynamics” investigate how traps affect hot-carrier relaxation in lead-halide perovskite materials. Using ultrafast transient absorption spectroscopy and photoluminescence measurements, they show that the hot carriers are directly captured by traps, instead of going through an intermediate cold carrier, addressing current important disunion within the community. The authors believe that these results will be important for future design of materials for hot-carrier solar cells. And I agree with them.

The manuscript is exceptionally well written with nice and clear figures illustrating the results of the work. I’m convinced that the conclusion, that the hot carriers are dependent on trap depth, is well proven and supported by data. With that, I would recommend the manuscript to be published in Nature Communications with minor revisions.

We are grateful to the reviewer for their time in evaluating our paper, and delighted to note that our careful experiments and analyses are convincing. We are also glad to see the reviewer

recognise the strong importance of these results to the wider community. All comments raised have now been addressed, as detailed below.

Prior to the possible publication, I would like the authors to address the following issues:

1) All the TA figures in the manuscript are plotted in wavelengths (nm) and “ $\Delta T/T$ ” while in the SI most of the TA data are plotted in energy scale (eV) and as “ $-\Delta T/T$ ” and only the last Supplementary Fig.11 is again in wavelength. While I do appreciate using different scales for different purposes, in this case it is really confusing, for someone trying to understand the observed dynamics. Is there any specific reason for presenting the same datasets in such a different way? If not, these should be unified, to avoid further confusion.

Response 2.1: We apologise for the confusion. We have now corrected all vertical axes to $\Delta T/T$ in Fig.2a-e, Fig.3a-l and Supplementary Fig. 8-10, and consistently use wavelength for the figures with TA spectra in the main text and SI.

2) Following the TA spectral evolution for better understanding the kinetics in Fig.2 (c, d, e) is quite difficult already, as mentioned above, however for the panel e (kinetics of CsPbI₃) it is rather impossible. The kinetics are plotted at 750-760 nm, but the corresponding TA data sets (both the matrix [a, b c] and differential spectra [d, e, f]) in the SI are shown only up to ~710 nm (1.75 eV). Thus, following the spectral evolution at that wavelength is, for the reader, impossible. The data should be shown.

Response 2.2: We thank the reviewer for pointing this out. We have now revisited the deconvolution of the TA spectra into the trap bleach and photoinduced absorption (PIA) components. This singular-value decomposition (SVD) is shown in Supplementary Fig. 7. We have now changed the wavelength range SVD is performed over for the CsPbI₃ series to 715–725 nm in order to coincide with the range of wavelengths measured and displayed in Supplementary Fig. 10d-f. We have checked to ensure that the trap bleach kinetics shown for CsPbBr₃ and CsPbBr_xI_{3-x} coincide with the wavelength range shown in the TA spectra for these materials (Supplementary Fig. 8 and 9).

Copied below are the trap bleach kinetics (Fig. 2e), SVD deconvolution (Supplementary Fig. 7c-d) and original TA spectra (Supplementary Fig. 10d and f) for CsPbI₃ NCs.

Fig. 1 | Evidence of carrier trapping from pump-probe transient absorption spectroscopy. *a, b*, Transient absorption spectra of the singly-purified (low defect density) and doubly-purified CsPbBr₃ NCs (high defect density). Red and blue regions indicate the ground state bleach (GSB) (490-500 nm) and sub-bandgap trap bleach (TB) (535-545 nm) regions that were integrated to determine the kinetics. The kinetics of the TB for singly (low defect density)- and double-purified (high defect density) *c*, CsPbBr₃ NCs (probed at 535-545 nm), *d*, CsPbBr_xI_{3-x} NCs (probed at 635-645 nm), and *e*, CsPbI₃ NCs (probed at 715-725 nm). The TA measurements were performed at 116.6 μJ cm⁻² pulse⁻¹ fluence, and the spectra are shown in Supplementary Fig. 8-10. TB was extracted from the measured spectra through singular-variable decomposition (SVD), shown in Supplementary Fig. 7. *f*. Schematic representation of the charge-carrier relaxation processes highlighting the carrier trapping events, with energy levels indicated here are arbitrarily. The NC solution was pumped with photon energies significantly higher than the bandgap ($\hbar\omega_{\text{pump}} = 3.1$ eV).

Supplementary Fig. 7 | Short-time transient absorption signal decomposition. Spectra deconvolution for *a*, low defects CsPbBr_xI_{3-x} NCs and *b*, high defects CsPbBr_xI_{3-x} NCs. Spectra deconvolution for *c*, low defects CsPbI₃ NCs and *d*, high defects CsPbI₃ NCs. The decomposition method is used in our previous report²⁵.

Supplementary Fig. 10 | Hot carrier cooling kinetics based on pump-probe transient absorption spectroscopy for CsPbI₃ NCs. *a-c*, TA maps, *d-f*, TA spectra and *g-i*, extracted charge-carrier temperatures vs. pump-probe delay of *a, d, h*, pristine (low defect density), *b, e, i*, single-purified (moderate defect density), and *c, f, g*, doubly-purified (high defect density) CsPbBr₃ perovskite nanocrystal solutions. The maps and spectra were recorded under 193.36 $\mu\text{J cm}^{-2}$ pulse⁻¹ fluence.

3) Throughout the manuscript, when describing the measured lifetimes, the authors write the values and their errors down to two decimals (for example 2.63 ± 0.41 ps). This, in fact means, that the lifetimes were determined with a precision down to tens of femtoseconds. However, I highly doubt this! In the Methods section, the laser used for TA is described as generating ~ 100 fs pulses, which together with ultrafast artifacts like GVD (chirp) and cross-phase modulation typically generates response function of at least ~ 150 fs, unless special precautions are made. This is even more striking in the SI, where the Fig.11 even shows lifetimes determined down to third decimal, thus with a precision down to a single femtosecond! I'm convinced, these numbers are just an artifact of the fitting procedure and that they have no physical (nor chemical) meaning as the data (and even the method) simply does not allow such a precision. These numbers should be rounded throughout the manuscript to a single decimal; hundreds of femtoseconds is a reasonable precision in this configuration, I'm convinced.

Response 2.3: This is a good point raised by the reviewer. The laser system we used is a Ti-Sapphire based ultrafast amplifier (~ 100 fs, repetition rate of 1000 Hz) modulated to 500 Hz through a chopper. The measurement step that we used at early times is 50 fs to give incremental observations of the carrier accumulation and cooling processes. We agree that the resolution of the system cannot reach tens of femtoseconds (*i.e.*, two decimal places). Rounding to 2 d.p. has been widely adopted in the community for reporting hot carrier cooling lifetimes, and we initially wished to be consistent with the wider community. For example, Pullerits *et al.* report 0.39 ps cooling time for CsPbBr₃ NCs, 0.27 ps for MAPbBr₃ NCs and 0.21 ps for FAPbBr₃ NCs²⁶, while Liu *et al.* reported 0.74 ps and 0.22 ps for CsPbBr₃ NCs with

different surface ligands²⁷. However, to avoid confusion over the precision of our measurements, we added the following clarification to the captions of Supplementary Tables 2-4 and Supplementary Fig. 12 to clarify that the decimal place is from fitting, not from measurement precision:

“The number of decimal places quoted for the mean values and uncertainties were obtained based on numerical fitting of the carrier temperature vs pump-probe delay time. We rounded to 1 d.p. to not imply that we can measure with better than 100 fs accuracy.”

Also, we have now corrected Supplementary Tables 2-4 to show lifetimes to 1 decimal place, and corrected to 3 d.p. to 2 d.p. in Supplementary Fig. 12.

Main text Page 10.

“the HC cooling lifetime decreases from 3.8 ± 0.3 ps (for the low defect density sample), to 2.6 ± 0.4 ps and 1.0 ± 0.1 ps”

“The cooling lifetime decreased from 7.1 ± 0.1 ps (for the low defect density sample) to 4.8 ± 0.2 ps to 4.5 ± 0.1 ps for the moderate to high defect density sample”

Supplementary Fig. 12 | Kinetics of GSB and PAs for the representative case of CsPb(Br/I)₃ NCs under PP and PPP-TA measurements. All samples are colloidal solutions, and are measured inside a 1 mm thick cuvette. The NIR push beam has an energy of 0.95 eV (1300 nm). The pump laser is a 400 nm pulse laser at $2 \mu\text{J cm}^{-2}$. *The number of decimal places quoted for the mean values and uncertainties were obtained based on numerical fitting of the carrier temperature vs pump-probe delay time.*

4) I really like the utilization of the 3 pulse TA in this case, the message it delivers is solid! The simple kinetic model can also quite surprisingly well describe the measured data. Only in one case the fit differs quite significantly from the data, which is rather surprising. Could the authors comment on why they think the measured $\tau(\text{hot})$ for CsPb(Br/I)₃ are lower than the model would predict?

Response 2.4: We appreciate very much the reviewer's positive remarks regarding our use of three-pulse measurements and the kinetic model we developed.

We revisited the fitting to improve the accuracy of the fit. We find that increasing the total phonon density (N_{ph}) for the mixed-halide NCs in the kinetic model brings better agreement with experimental results. This is physically meaningful as the breaking of lattice symmetry caused by the mixing halides ought to introduce additional phonon modes to the system. This is analogous to previous work by Hopper *et al.*²⁸, wherein the hot phonon bottleneck was suppressed in perovskites containing a molecular cation over that of their monatomic Cs-containing counterparts. We have updated the quoted fitting parameters in Supplementary Table 5. We have also corrected an error from the original submission where the units for N_{ph} in this Table were written as $\times 10^{12} \text{ cm}^{-2}$ instead of the intended $\times 10^{11} \text{ cm}^{-2}$. The improved results are shown here, and also in main text Fig. 5e and Supplementary Table 5:

Fig. 5e HC lifetimes obtained from fitting the PPP TA measurements for **e**, CsPbBr_xI_{3-x} NCs. Solid lines are the results from the kinetic model described later.

Supplementary Table 5 | Fitting parameters for a numerical kinetic model describing hot carrier dynamics shown in Fig. 5 d-f.

Fit parameter (unit)		CsPbBr ₃	CsPb(Br/I) ₃	CsPbI ₃
Pump carrier density	N_{pump} ($\times 10^{12} \text{ cm}^{-2}$)	1.1	1.0	1.0
Hot-cold carrier scattering	α ($\times 10^{-12} \text{ cm}^2 \text{ ps}^{-1}$)	1.3	1.3	1.3
Carrier-phonon scattering	ϕ ($\times 10^{-12} \text{ cm}^2 \text{ ps}^{-1}$)	7.0	3.5	3.5
Carrier-trap scattering	γ ($\times 10^{-12} \text{ cm}^2 \text{ ps}^{-1}$)	1.5	1.5	0.0
Phonon freeing	η (ps^{-1})	0.15	0.15	0.03
Total phonon density	N_{ph} ($\times 10^{11} \text{ cm}^{-2}$)	8.0	9.5	5.5

5) The Supplementary Fig. 1 has two panels “e”.

Response 2.5: We thank the reviewer for pointing this out this, we have corrected the labelling in this figure, as well as in the caption

Supplementary Fig. 2 | Transmission Electron Microscopy. **a-c**, TEM images for pristine (low defect density), single-purified (moderate defect density), and doubly-purified (high defect density) CsPbBr_3 perovskite nanocrystal (NCs). **d-f**, TEM images for pristine (low defect density), single-purified (moderate defect density), and doubly-purified (High Defects) $\text{CsPbBr}_x\text{I}_{3-x}$ NCs. **g-i**, TEM images for pristine (low defect density), single-purified (moderate defect density), and doubly-purified (high defect density) CsPbI_3 NCs.

Reviewer 3:

The manuscript by Hoye and co-workers reports a study of hot-carrier cooling dynamics in CsPbBr_3 , CsPbI_3 , and $\text{CsPb}(\text{Br/I})_3$ NCs with femtosecond pump-probe and pump-push-probe transient absorption spectroscopy. The NCs were washed to deliberately create traps to investigate the effect of traps on the hot carrier cooling process in the three systems. The authors find that the decay process follows the defect-tolerance nature of the NCs, like that of band edge carrier decay. They found that the hot carriers can be directly captured by traps before going through intermediate cold carriers, and thus deeper traps lead to faster cooling.

The kinetic modeling used to explain the hot carrier cooling lifetimes fits well with the experimental data. The experiments are highly innovative, and the results are very interesting. Therefore, the manuscript can be interesting for publication in Nature Communications. I recommend to publish it. But a few issues need the author's attention.

We are delighted to see the keen interest of the reviewer in our work, and their strongly positive appraisal of the innovative nature of our experiments. All comments raised have now been addressed, as detailed below.

1. Could the author explain briefly in the text how they calculate the carrier density shown in Figure 4?

Response 3.1: The carrier density estimation is made using a previously-reported method²⁹. We have now elaborated on the details of this in the Methods section on page 17:

“The carrier density was calculated using the method reported in Ref. 49. Based on the Poisson distribution of the initial photon occupancies in NCs, the probability of one NC containing i number of excitons is given by $p_i = \left(\frac{\langle N \rangle^i}{i!}\right) e^{-\langle N \rangle}$, where N is the average number of excitons per nanocrystals, $\langle N \rangle = \sigma j$, p_i the probability of a nanocrystal containing i excitons, j is the pump fluence, and σ is the absorption cross-section of the nanocrystals. TA signal intensity is directly proportional to σj when there is less than 1 exciton per NCs, i.e., $I_{TA} \propto 1 - e^{-\langle N \rangle} = 1 - e^{-\sigma j}$. The estimated σ for Br-based NCs is $3.1 \times 10^{-15} \text{ cm}^2$, for mixed Br/I NCs is $4.2 \times 10^{-15} \text{ cm}^2$, and for I-based NCs is $6.2 \times 10^{-15} \text{ cm}^2$. The carrier density, n_0 is estimated based on the volume of the NCs obtained by TEM, $n_0 = \frac{\langle N \rangle}{V_{NC}}$.”

2. What is the fitting window to obtain the temperature vs time graph shown in Supplementary Fig. 8-10? Could the author justify their choice of selecting the fitting windows?

Response 3.2: The fitting range is now indicated by the dashed lines added to Supplementary Fig. 8-10. For consistency, the fitting window is maintained at 0.2 eV. This was chosen in order to cover the high energy part of the TA spectra that is due to hot carrier cooling, whilst not including the decay from the GSB.

To the captions of Supplementary Fig. 8–10, we have added the following:

“In d-f, the fitting window for obtaining hot carrier cooling kinetics is indicated with the dashed lines, and was chosen to cover the relevant energy range for hot carriers, whilst not being affected by the decay in the GSB.”

3. In Fig.3j, the maximum carrier temperatures for samples with different defects are different, the low defects sample can reach almost 4000K, while the high defect sample only 1000K, could the author explain?

Response 3.3: We believe that the maximum temperature difference between the high and low defect density NC samples is due to a decrease in the strength of the defect-hot carrier coupling strength (H) when the defect density is low. This is supported by the fitting shown in Supplementary Table 1, where the radiative constant k_2 is higher and constant b (related to defect-carrier coupling) is lower for the low defect density samples. The low coupling constant blocks relaxation pathways through defects in the first few hundreds of femtoseconds, leading to a higher early-time carrier temperature (maximum temperature). Ideally, the carrier

temperature (at early time) of similar band gap materials for a given excitation energy should be the same due to same amount of excess energy. However, in our case, under the same experimental conditions, the high defect density samples produced lower initial carrier temperatures possibly due defect-induced energy loss channels during the early thermalisation process³⁰. An abundance of defect sites would introduce an efficient carrier-defect scattering channel, leading to rapid energy losses even before the instrument response time³⁰.

4. Why do the CsPbI₃ nanocrystals degrade after two washing cycles?

Response 3.4: The nature of the instability of CsPbI₃ NCs comes from the phase sensitivity of the material due to lattice distortion and octahedral tilting of the Pb-I framework. The stable α -phase (cubic) can be maintained for bulk CsPbI₃ at high temperature, or at room temperature in the form of NCs with good surface ligand coverage to avoid large distortions and prevent the formation of the metastable β -phase (tetragonal) or photo-inactive γ -phase (orthorhombic). However, as the density of ligands reduces during washing with polar antisolvents, the NC surface become less stable and more likely to form γ -phase or degradation as I-ions are very sensible to moisture or polar solvents exposure^{18,31,32}. Therefore, the doubly-washed CsPbI₃ NCs become less stable when stored under ambient conditions.

5. Is the model used to fit the PPP cooling data also capable of describing the PP cooling data?

Response 3.5: In the two-pulse PP approach described thus far, all charge-carriers (hot and cold) are formed by a single ('pump') excitation event. As such, the density-dependence of the intrinsic HC lifetime (including the effect of trapping) can be obscured by the 'Auger re-heating' effects at high excitation fluences. Moreover, in PP-TAS, the HCs are initially formed with different excess energies due to the variations in band gap between different NC compositions. Unfortunately, it would be difficult to disentangle the carrier-carrier, carrier-phonon, carrier-defect interactions from pump-probe data in PP. This contrasts to PPP experiments, in which the separate pump and push pulses allows independent control of the cold and hot-carrier densities, respectively. For a reasonably low cold carrier density, increase of push-fluence allows us to identify the role of hot-carrier-phonon interactions on hot-carrier cooling dynamics. In PP experiments, deconvolution of hot and cold-carriers' densities are not possible as both are governed by the action of pump-pulse alone. Therefore, the same model cannot be simply applied to fit the PP data.

6. Why the authors didn't compare the results with CsPbCl₃ NCs, which are less defect-tolerant?

Response 3.6: We did not involve Cl-based NCs due to technical difficulties. Firstly, the large band gap necessitates excitation pump laser wavelengths shorter than 400 nm, which is beyond our current experimental capabilities. Although in the PPP experimental setup, we could excite below 400 nm, our probing window was not able to cover the wide bandgap of Cl-based NCs, *i.e.*, the white light probe covers from 450 nm only.

Furthermore, CsPbCl₃ NCs often possess a wide range of recombination centres limiting its PLQY often below 10-15% (*i.e.*, Cl-based perovskites are not defect tolerant to cold carriers), hence it would be difficult to compare to the rest of the halide perovskite NCs, which exhibit relatively large PLQY in the range of 60-80%. Therefore, investigation of how defects influence hot carrier cooling dynamics in CsPbCl₃ goes beyond the scope of the present work.

7. Did the authors try to passivate and then see if the decay dynamics recover to the original?

Response 3.7: We could consider passivation as another method to control the NC defect density (in addition to the polar antisolvent washing method we used herein). We did not investigate the effect of passivation on hot carrier cooling kinetics, but other studies have shown similar phenomena, where passivating NCs can prolong hot carrier cooling lifetimes, and reduce the effect of hot carrier trapping. For example, Dai *et al.* showed by passivating CsSn_xPb_{1-x}I₃ NCs with Na ions, the hot carrier cooling lifetime could be increased³⁰. Righetto *et al.* passivated MAPbBr₃ NCs with trioctylphenylphosphine (TOPO) ligands to reduce the PLQY reduction when pumping with higher energy laser, which implies less hot carrier trapping effect after passivation¹⁹. We did not perform this passivation investigation because we did not wish to alter the surface ligand types, as Liu *et al.* observed that replacing oleylamine ligands with (3-aminopropyl)triethoxysilane (APTES) ligands on CsPbBr₃ NC surface can also alter the hot carrier cooling kinetics²⁷. Therefore, it is a cleaner experiment to keep the ligand types the same, and to reduce their density around the NCs (and introduce a higher defect density) as we did here.

8. Authors might consider referring to relevant articles like Nano Lett. 2017, 17, 9, 5402–5407; J. Phys. Chem. Lett. 2021, 12, 36, 8732–8739.

Response 3.8: We thank the reviewer for these suggestions, and have added them to the following places in the main text,

J. Phys. Chem. Lett. 2021, 12, 36, 8732–8739 is added here to show that pressure induced lattice compression can also influence HC lifetime.

Page 3: *HC-activated trapping*^{27,28}, along with *passivation-induced enhancement of HC lifetime*^{9,15,19,28-31}, *pressure-induced lattice compression*³² and *transport*³³, was also observed in a few other instances.

Nano Lett. 2017, 17, 9, 5402–5407 is added here as it shows extraction of hot carriers by probing the hot electron and hot hole lifetime with Terahertz Spectroscopic after connecting NC films with different transport layer.

Page 4: *also its wide applications in display, optical communication and energy harvesting*³⁷⁻⁴⁰

References

- 1 Otero-Martínez, C. *et al.* Organic A-Site Cations Improve the Resilience of Inorganic Lead-Halide Perovskite Nanocrystals to Surface Defect Formation. *Advanced Functional Materials* 2404399 (2024). <https://doi.org/https://doi.org/10.1002/adfm.202404399>
- 2 Nenon, D. P. *et al.* Design Principles for Trap-Free CsPbX₃ Nanocrystals: Enumerating and Eliminating Surface Halide Vacancies with Softer Lewis Bases. *J. Am. Chem. Soc.* **140**, 17760-17772 (2018). <https://doi.org/10.1021/jacs.8b11035>
- 3 Yin, W.-J., Shi, T. & Yan, Y. Unusual defect physics in CH₃NH₃PbI₃ perovskite solar cell absorber. *Appl. Phys. Lett.* **104**, 063903 (2014). <https://doi.org/10.1063/1.4864778>
- 4 Brandt, R. E., Stevanović, V., Ginley, D. S. & Buonassisi, T. Identifying defect-tolerant semiconductors with high minority-carrier lifetimes: beyond hybrid lead

- halide perovskites. *MRS Communications* **5**, 265-275 (2015).
<https://doi.org/10.1557/mrc.2015.26>
- 5 Zheng, X. *et al.* Reducing Defects in Halide Perovskite Nanocrystals for Light-Emitting Applications. *The Journal of Physical Chemistry Letters* **10**, 2629-2640 (2019). <https://doi.org/10.1021/acs.jpcllett.9b00689>
- 6 Song, H. *et al.* A Universal Perovskite Nanocrystal Ink for High-Performance Optoelectronic Devices. *Advanced Materials* **35**, 2209486 (2023).
<https://doi.org/https://doi.org/10.1002/adma.202209486>
- 7 Li, C. *et al.* Multifunctional ligand-manipulated luminescence and electric transport of CsPbI₃ perovskite nanocrystals for red light-emitting diodes. *Chemical Engineering Journal* **493**, 152483 (2024).
<https://doi.org/https://doi.org/10.1016/j.cej.2024.152483>
- 8 Wang, Z. *et al.* Superwetting Nanofluids of NiO_x-Nanocrystals/CsBr Solution for Fabricating Quality NiO_x-CsPbBr₃ Gradient Hybrid Film in Carbon-Based Perovskite Solar Cells. *Small Methods* 2400283 (2024).
<https://doi.org/https://doi.org/10.1002/smt.202400283>
- 9 Kim, D. *et al.* Investigation of potassium doping and defect healing mechanism in core-shell CsPbBr₃/SiO₂ quantum dots. *Journal of Materials Chemistry C* **12**, 6395-6405 (2024). <https://doi.org/10.1039/D4TC00537F>
- 10 Le Corre, V. M. *et al.* Revealing Charge Carrier Mobility and Defect Densities in Metal Halide Perovskites via Space-Charge-Limited Current Measurements. *ACS Energy Letters* **6**, 1087-1094 (2021).
<https://doi.org/10.1021/acsenerylett.0c02599>
- 11 Chen, Z. *et al.* Synergistic crystallization regulation and defect passivation for growth of high-quality perovskite single crystals towards ultrasensitive X-ray detection. *Journal of Materials Chemistry A* (2024).
<https://doi.org/10.1039/D4TA01688B>
- 12 Bechir, M. B. & Alresheedi, F. Growth methods' effect on the physical characteristics of CsPbBr₃ single crystal. *Physical Chemistry Chemical Physics* **26**, 1274-1283 (2024). <https://doi.org/10.1039/D3CP04645A>
- 13 Fang, Z. *et al.* Dual Passivation of Perovskite Defects for Light-Emitting Diodes with External Quantum Efficiency Exceeding 20%. *Adv. Funct. Mater.* **30**, 1909754 (2020). <https://doi.org/https://doi.org/10.1002/adfm.201909754>
- 14 Li, D. *et al.* Uniaxial-Oriented Perovskite Films with Controllable Orientation. *Advanced Science* **11**, 2401184 (2024).
<https://doi.org/https://doi.org/10.1002/advs.202401184>
- 15 Zhang, J. *et al.* CsPbBr₃ Nanocrystal Induced Bilateral Interface Modification for Efficient Planar Perovskite Solar Cells. *Advanced Science* **8**, 2102648 (2021).
<https://doi.org/https://doi.org/10.1002/advs.202102648>
- 16 Ma, L. *et al.* Eliminating Chlorine Vacancies of Perovskite Nanocrystals Using Hydrazine Cations Enables Efficient Pure Blue Light-Emitting Diodes. *ACS Energy Letters* **9**, 1210-1218 (2024). <https://doi.org/10.1021/acsenerylett.4c00109>
- 17 Gao, L. *et al.* Color-Stable and High-Efficiency Blue Perovskite Nanocrystal Light-Emitting Diodes via Monovalent Copper Ion Lowering Lead Defects. *ACS Applied Materials & Interfaces* **13**, 55380-55390 (2021).
<https://doi.org/10.1021/acsam.1c18041>

- 18 Ye, J. *et al.* Elucidating the Role of Antisolvents on the Surface Chemistry and Optoelectronic Properties of CsPbBr_{3-x} Perovskite Nanocrystals. *J. Am. Chem. Soc.* **144**, 12102-12115 (2022). <https://doi.org/10.1021/jacs.2c02631>
- 19 Righetto, M. *et al.* Hot carriers perspective on the nature of traps in perovskites. *Nat. Commun.* **11**, 2712 (2020). <https://doi.org/10.1038/s41467-020-16463-7>
- 20 Otero-Martínez, C. *et al.* Fast A-Site Cation Cross-Exchange at Room Temperature: Single-to Double- and Triple-Cation Halide Perovskite Nanocrystals. *Angewandte Chemie International Edition* **61**, e202205617 (2022). <https://doi.org/https://doi.org/10.1002/anie.202205617>
- 21 Ye, J. *et al.* Strongly-confined colloidal lead-halide perovskite quantum dots: from synthesis to applications. *Chemical Society Reviews* (2024). <https://doi.org/10.1039/D4CS00077C>
- 22 Ye, J. *et al.* Direct linearly polarized electroluminescence from perovskite nanoplatelet superlattices. *Nature Photonics* **18**, 586-594 (2024). <https://doi.org/10.1038/s41566-024-01398-y>
- 23 Ren, A. *et al.* High-bandwidth perovskite photonic sources on silicon. *Nature Photonics* **17**, 798-805 (2023). <https://doi.org/10.1038/s41566-023-01242-9>
- 24 Sarkar, S. *et al.* Terahertz Spectroscopic Probe of Hot Electron and Hole Transfer from Colloidal CsPbBr₃ Perovskite Nanocrystals. *Nano Letters* **17**, 5402-5407 (2017). <https://doi.org/10.1021/acs.nanolett.7b02003>
- 25 Huang, Y.-T. *et al.* Strong absorption and ultrafast localisation in NaBiS₂ nanocrystals with slow charge-carrier recombination. *Nat. Commun.* **13**, 4960 (2022). <https://doi.org/10.1038/s41467-022-32669-3>
- 26 Chen, J., Messing, M. E., Zheng, K. & Pullerits, T. Cation-Dependent Hot Carrier Cooling in Halide Perovskite Nanocrystals. *J. Am. Chem. Soc.* **141**, 3532-3540 (2019). <https://doi.org/10.1021/jacs.8b11867>
- 27 Zeng, P. *et al.* Control of Hot Carrier Relaxation in CsPbBr₃ Nanocrystals Using Damping Ligands. *Angewandte Chemie International Edition* **61**, e202111443 (2022). <https://doi.org/https://doi.org/10.1002/anie.202111443>
- 28 Hopper, T. R. *et al.* Ultrafast Intraband Spectroscopy of Hot-Carrier Cooling in Lead-Halide Perovskites. *ACS Energy Letters* **3**, 2199-2205 (2018). <https://doi.org/10.1021/acsenerylett.8b01227>
- 29 Dai, L. *et al.* Slow carrier relaxation in tin-based perovskite nanocrystals. *Nature Photonics* **15**, 696-702 (2021). <https://doi.org/10.1038/s41566-021-00847-2>
- 30 Dai, L., Ye, J. & Greenham, N. C. Thermalization and relaxation mediated by phonon management in tin-lead perovskites. *Light: Science & Applications* **12**, 208 (2023). <https://doi.org/10.1038/s41377-023-01236-w>
- 31 Dutta, A. & Pradhan, N. Phase-Stable Red-Emitting CsPbI₃ Nanocrystals: Successes and Challenges. *ACS Energy Letters* **4**, 709-719 (2019). <https://doi.org/10.1021/acsenerylett.9b00138>
- 32 Sun, J.-K. *et al.* Polar Solvent Induced Lattice Distortion of Cubic CsPbI₃ Nanocubes and Hierarchical Self-Assembly into Orthorhombic Single-Crystalline Nanowires. *J. Am. Chem. Soc.* **140**, 11705-11715 (2018). <https://doi.org/10.1021/jacs.8b05949>

Reviewer #1 (Remarks to the Author):

The authors have answered all my questions and this work has met the requirements for acceptance.

Reviewer #2 (Remarks to the Author):

The authors managed to address all the issues I have pointed out, thus I do recommend the manuscript for publication.

Reviewer #3 (Remarks to the Author):

The authors have thoroughly revised the manuscript. They have satisfactorily addressed the previous comments and suggestions. I recommend to publish the present version in Nature Communications.